



# Anthropogenic CO₂ emission estimates in the Tokyo Metropolitan Area from ground-based CO₂ column observations

Hirofumi Ohyama[1,*], Matthias M. Frey[1,*], Isamu Morino[1], Kei Shiomi[2], Masahide Nishihashi[1,a], Tatsuya Miyauchi[1,b], Hiroko Yamada[1], Makoto Saito[1], Masanobu Wakasa[3], Thomas Blumenstock[4], Frank Hase[4]

[1] Earth System Division, National Institute for Environmental Studies, Tsukuba, Japan
[2] Earth Observation Research Center, Japan Aerospace Exploration Agency, Tsukuba, Japan
[3] Graduate School of Science and Engineering, Saitama University, Saitama, Japan
[4] Karlsruhe Institute of Technology (KIT), Institute of Meteorology and Climate Research (IMK-ASF), Karlsruhe, Germany
[a] now at Department of Observation and Data Assimilation Research, Meteorological Research Institute, Tsukuba, Japan
[b] now at Research Faculty of Agriculture, Hokkaido University, Sapporo, Japan
*These authors contributed equally to this work.

*Correspondence to*: Hirofumi Ohyama (oyama.hirofumi@nies.go.jp) and Matthias M. Frey (frey.matthias.max@nies.go.jp)

**Abstract.** Urban areas are responsible for more than 40 % of global energy-related carbon dioxide (CO₂) emissions. The Tokyo Metropolitan Area (TMA), Japan, one of the most populated regions in the world, includes various emission sources,

such as thermal power plants, automobile traffic, and residential facilities. We conducted an intensive field campaign in the TMA from February to April 2016 to measure column-averaged dry-air mole fractions of CO₂ (XCO₂) with three ground-based Fourier transform spectrometers (one IFS 125HR and two EM27/SUN spectrometers). At two urban sites (Saitama and Sodegaura), measured XCO₂ values were generally larger than those at a rural site (Tsukuba) by up to 9.5 ppm, and average diurnal variations increased toward evening. To simulate the XCO₂ enhancement (ΔXCO₂) resulting from emissions at each

observation site, we used the Stochastic Time-Inverted Lagrangian Transport (STILT) model driven by meteorological fields at a horizontal resolution of ~1 km from the Weather Research Forecast (WRF) model, which was coupled with anthropogenic (large point source and nonpoint source) CO₂ emissions and biogenic fluxes. Although some of the diurnal variation of ΔXCO₂ was not reproduced and plumes from nearby large point sources were not captured, primarily because of a transport modeling error, the WRF–STILT simulations using prior fluxes were generally in good agreement with the observations (mean bias,

0.30 ppm; standard deviation, 1.31 ppm). By combining observations with high-resolution modeling, we developed an urban-scale inversion system in which spatially resolved CO₂ emission fluxes at >3 km resolution and a scaling factor of large point source emissions were estimated on a monthly basis by using Bayesian inference. The ΔXCO₂ simulation results from the posterior CO₂ fluxes were improved (mean bias, –0.03 ppm; standard deviation, 1.21 ppm). In addition, the inverse analysis reduced the uncertainty in total CO₂ emissions in the TMA by a factor of ~2. The posterior total CO₂ emissions agreed with

emission inventories within the posterior uncertainty at the 95 % confidence level, demonstrating that the EM27/SUN spectrometer data can constrain urban-scale monthly CO₂ emissions.



## 1 Introduction

The steady increase of atmospheric greenhouse gas (GHG) concentrations has accelerated recent climate change. Although urban areas account for only 2–3 % of the global land surface, approximately 44 % of energy-related carbon dioxide ($CO_2$) emissions come directly from urban areas (Seto et al., 2014). Urban areas are thus a main target for emission reductions, and many cities around the world have committed to reduce their GHG emissions through both the C40 Cities Climate Leadership Group (https://www.c40.org/) and city-specific programs. To support urban emission reduction strategies, a variety of efforts

are currently underway to build emission inventories with high spatial and temporal resolution. These inventories estimate GHG emissions by using a bottom-up approach, in which the total emissions from each source category are calculated by multiplying activity data (e.g., fuel consumption, traffic counts, housing statistics) by emission factors indicating GHG emissions per unit of activity. Because such detailed inventories have been developed only for certain cities, another type of global emissions database has been developed that relies on spatial proxies (e.g., night lights, population) to downscale total

emissions at national or sub-national scales. Gurney et al. (2019), however, have reported large discrepancies between downscaled and bottom-up fossil fuel $CO_2$ emissions at the urban scale, largely due to large point sources and road traffic. Independent verification of these emissions datasets is highly desirable, and atmospheric observations are increasingly being used for this purpose. Emissions can be estimated at fine scale from atmospheric observations of GHG concentrations by using high-resolution atmospheric transport models in a top-down inversion approach. Networks of in situ GHG observation stations

providing both operational observations and emission estimates have been established in several megacities (e.g., McKain et al., 2012; Lian et al., 2022; Lauvaux et al. 2016; Sargent et al., 2018). In addition, emission estimates have been obtained by conducting aircraft-based measurement campaigns over megacities (Ahn et al., 2020; Pitt et al., 2022) and by using laser absorption spectroscopy for 2-D tomographic measurements (Lian et al., 2019). For the Tokyo Metropolitan Area (TMA), Japan, the most populous metropolitan area in the world, Pisso et al. (2019) estimated anthropogenic $CO_2$ fluxes during the

winters of 2005–2009 at a spatial resolution of 20 km × 20 km from in situ measurements obtained by tower- or ground-based instruments (Tsukuba, Kisai, and Dodaira) and commercial aircraft-based instruments (over Narita) by an inverse analysis with a Lagrangian transport model. Recently, the number of tower- and ground-based observation sites in the TMA have been expanded, and additional atmospheric components and isotopes are being measured (Sugawara et al., 2021).

    Compared with surface point measurements, total column measurements are less sensitive to changes in the height of the

planetary boundary layer (PBL) (Olsen and Randerson, 2004; McKain et al., 2012). Therefore, column measurements help to both mitigate PBL height errors in an atmospheric inversion system (Gerbig et al., 2008) and disentangle the effects of atmospheric mixing from the exchange of carbon between the surface and the atmosphere (Wunch et al., 2011). In addition, column data obtained in urban areas include information about emissions over a broader spatial domain than surface point data. Babenhauserheide et al. (2020) have estimated $CO_2$ emissions from Tokyo by conducting a statistical analysis of long-

term measurements of column-averaged dry-air $CO_2$ mole fractions ($XCO_2$) obtained with a ground-based high-resolution Fourier transform spectrometer (FTS, IFS 125HR, Bruker Optics) located at Tsukuba, about 50 km away from Tokyo, together with wind data obtained from operational radiosonde observations. GHG emissions from several megacities have been characterized by field campaigns conducted with multiple portable FTSs (EM27/SUN, Bruker Optics) (e.g., Hase et al., 2015; Makarova et al., 2021). Comparison of observed $XCO_2$ and column-averaged dry-air methane mole fraction ($XCH_4$) values

with simulation results obtained by high-resolution transport modeling demonstrated that the simulations could capture $XCO_2$ and $XCH_4$ gradients between upwind and downwind sites (Vogel et al., 2019; Zhao et al., 2019, 2022). Furthermore, emissions from megacities have been estimated from $XCO_2$ and $XCH_4$ data with Lagrangian transport models (Ionov et al., 2021; Jones et al., 2021; Hedelius et al., 2018). Cusworth et al. (2020) have estimated spatially resolved $CH_4$ emissions in the Los Angeles basin at a resolution of 3 km × 3 km from operational surface and column data. Meanwhile, permanent city observatories with

the EM27/SUN spectrometers are emerging (e.g., Dietrich et al., 2021).

    In the present study, we performed an observation campaign using two EM27/SUN FTSs and the Tsukuba IFS 125HR FTS to



constrain $CO_2$ emissions around Tokyo during late winter and early spring, when the daily sunshine duration in this region is high and the contribution of the biogenic flux to $CO_2$ fluctuations is minor. To simulate atmospheric transport at high spatiotemporal resolution, we used a Lagrangian transport model, the Stochastic Time-Inverted Linear Transport (STILT) model, driven by the Weather Research and Forecasting (WRF) model (Lin et al., 2003; Nehrkorn et al., 2010). We estimated spatially resolved anthropogenic $CO_2$ emissions by an inverse analysis and then evaluated the estimated monthly $CO_2$ emissions against inventory-based fossil fuel $CO_2$ emissions in the TMA. In Sect. 2, we describe the $XCO_2$ measurements by ground-based FTSs at three observation sites in the TMA. Section 3 presents a methodology for modeling $XCO_2$ enhancements at the observation sites using the atmospheric transport model and prior emission data, and a framework for estimating anthropogenic $CO_2$ emissions through Bayesian inference. In Sect. 4, we show the results of the $XCO_2$ measurements and simulations and discuss the posterior emission estimates.

## 2 Measurements during the observation campaign in the TMA

An observation campaign with portable FTSs and the Tsukuba FTS (the 2016 Tokyo campaign) was conducted in the TMA from February to April 2016. Here, the TMA is defined as a rectangular region that includes the urban areas of Kanagawa, Chiba, Saitama, Ibaraki, Tochigi, and Gunma prefectures as well as the Tokyo metropolis (Fig. 1). The United Nations reports that "Tokyo", with approximately 38 million inhabitants, is the world's most populous area and accounts for 30.1 % of the total population of Japan (United Nations, 2018), although the boundaries used to define Tokyo by the United Nations are not the same as those used here. On the one hand, the city center of the TMA, primarily because of its intense economic activity and high population density, is a strong source of anthropogenic $CO_2$ emissions (Saito et al., 2023), and there are a multitude of large point sources, such as power plants and steel plants, along the shores of Tokyo Bay. On the other hand, the TMA is surrounded by evergreen broadleaf and deciduous broadleaf forests, which contribute to temporal variations in biogenic $CO_2$ fluxes. The 2016 Tokyo campaign was conducted from late winter to early spring, when biological activity within the TMA was dormant; thus, it was likely responsible for much smaller changes in $CO_2$ concentrations than the anthropogenic activity. We used measurement data from the ground-based high-resolution FTS operated as part of the Total Carbon Column Observing Network (TCCON, Wunch et al., 2011) at the National Institute for Environmental Studies (NIES) (35.0513° N, 140.1215° E, 31 m above sea level (ASL)) in Tsukuba (Morino et al., 2018). Additionally, we used data from two EM27/SUN spectrometers (SN38 and SN44) throughout the campaign period and a third EM27/SUN spectrometer (SN63) beginning in the middle of the campaign. Considering the prevailing wind direction in this winter/early spring (i.e., northwesterly) and proximity to anthropogenic emission sources, we deployed the SN38 EM27/SUN spectrometer at Saitama University (35.8636° N, 139.6081° E, 28 m ASL) and the SN44 EM27/SUN spectrometer at Sodegaura City Hall (35.4297° N, 139.9545° E, 14 m ASL) from 16 February 2016 to 6 April 2016 (Fig. 1). The SN63 EM27/SUN spectrometer began operation at Tsukuba on 3 March 2016 and is now continuously operated as part of the Collaborative Carbon Column Observing Network (COCCON, Frey et al., 2019). Before and after the observations at the three sites, we conducted side-by-side measurements with the EM27/SUN instruments and the TCCON instrument at NIES for approximately one week each time (3−10 February and 11−20 April).

The EM27/SUN instrument measures direct solar absorption spectra from 4000 to 11,000 $cm^{-1}$ in the short-wavelength infrared (SWIR) region (Gisi et al., 2012). At the time of the 2016 Tokyo campaign, the participating EM27/SUN spectrometers were equipped with only one InGaAs detector, and column amounts of $CO_2$, $CH_4$, water vapor, and oxygen were obtained from the SWIR spectra. The spectral resolution was 0.5 $cm^{-1}$, which corresponds to an optical path difference of 1.8 cm. Interferograms were continuously acquired every 6 s, and every 10 interferograms (five each from the forward and backward scans) were averaged and recorded (i.e., a sampling interval of approximately 1 min). If the weather permitted, EM27/SUN measurements at Saitama were made from sunrise to sunset, whereas the measurements at Sodegaura were made between approximately



09:00 Japan Standard Time (JST) and sunset, reflecting the office hours of the city hall. For retrieval processing, we used GGG2014 software, which is also used for processing the TCCON spectra (Wunch et al., 2015). To ensure consistency among

the instruments, we determined correction factors for $XCO_2$ and $XCH_4$ values based on the side-by-side measurements performed at NIES before and after the field campaign. The SN63 EM27/SUN data, which were bias-corrected against coincident aircraft measurements (Ohyama et al., 2020), were used as reference data. The instrumental line shape of the SN44 EM27/SUN deviated greatly from that of an ideal instrument because of its imperfect alignment (Frey et al., 2019); therefore, airmass-dependent correction factors (ADCFs) were derived in addition to airmass-independent correction factors (AICFs).

The procedure for determining these correction factors is described in detail in Ohyama et al. (2021). The resulting correction factors $C_0$ (AICF) and $C_1$ (ADCF) in Eq. (1) of Ohyama et al. (2021) were 1.0028 and 0.0096, respectively, for the SN44 EM27/SUN $XCO_2$ data. The $C_0$ value of the SN38 EM27/SUN $XCO_2$ data was 1.0006. The $C_0$ and $C_1$ values of SN44 EM27/SUN $XCH_4$ data were 1.0101 and 0.0021, respectively, and the $C_0$ value of the SN38 EM27/SUN $XCH_4$ data was 1.0017. Comparisons of the bias-corrected EM27/SUN data with the SN63 EM27/SUN data are shown in Fig. 2. After the

bias correction, the SN38 and SN44 EM27/SUN $XCO_2$ data differed from the SN63 EM27/SUN $XCO_2$ data by (mean ± $1\sigma$) $-0.01 \pm 0.17$ ppm and $0.06 \pm 0.16$ ppm, respectively, and the SN38 and SN44 EM27/SUN $XCH_4$ data differed from the SN63 EM27/SUN $XCH_4$ data by $-0.09 \pm 0.97$ ppb and $0.39 \pm 0.88$ ppb, respectively. In the present study, the Tsukuba TCCON data were also scaled to be consistent with the SN63 EM27/SUN data by using $C_0$ values of 0.9977 for $XCO_2$ and 0.9985 for $XCH_4$. Figure 3 shows time series of the bias-corrected $XCO_2$ and $XCH_4$ data observed with the four spectrometers during the

campaign period, including the side-by-side measurements at the Tsukuba site.

**3 Methodology for the $CO_2$ emission estimation**

**3.1 Lagrangian transport model**

We used the STILT model (Lin et al., 2003; Fasoli et al., 2018) coupled with meteorological fields from the WRF model (hereafter WRF–STILT; i.e., STILT driven by meteorological fields from WRF) to simulate atmospheric transport as required

for inversion of the $CO_2$ emission data. STILT calculates the trajectory of particles from a "receptor" location and generates a footprint that represents the sensitivity of the $CO_2$ mole fraction to be measured at the receptor location to upwind emissions. This footprint (concentration per unit flux; ppm $m^{-2}$ $s^{-1}$) corresponds to the Jacobian matrix used for inverse analysis of $CO_2$ emissions. We used WRF meteorological fields generated at a horizontal resolution of approximately 1 km to drive STILT (described in detail in Sect. 3.2). We ran the model at 14 receptor levels (25, 50, 75, 100, 150, 200, 300, 400, 600, 800, 1000,

1500, 2000, and 2800 m above ground level) over each observation site, and an ensemble of 1000 particles for each altitude was traced backwards in time for 24 h. We varied the location (latitude and longitude) of the receptor along the line-of-sight of the EM27/SUN pointing toward the sun to accord with the receptor level. The footprints for each grid were computed by considering the PBL height and the residence time of particles traveling within the lower PBL (Lin et al., 2003). The footprint calculations were performed every 15 min from 09:00 to 17:00 JST at a spatial resolution of 30 arcsec (approximately 1 km)

within 34.975° N–36.625° N and 138.900° E–140.875° E (Fig. 1). We then aggregated the footprints in each grid over the STILT run time.

From the STILT footprints at each altitude, we calculated the pressure-weighted column-average footprint, taking account of the column-averaging kernel of the EM27/SUN spectrometer (Rodgers and Connor, 2003; Jones et al., 2021). The footprints generated by STILT were then multiplied by anthropogenic and biogenic fluxes to determine the spatially resolved

contributions (in ppm) of the surrounding emission sources. The change in $XCO_2$ ($\Delta XCO_2$) at each observation site was obtained by summing the contributions over all grid cells. We separated the anthropogenic flux into large point source and nonpoint source emissions, as described in Sect. 3.3. We thus considered three types of fluxes: large point source emissions, nonpoint source emissions, and the biogenic flux.



### 3.2 Meteorological fields from WRF model

To drive the STILT model, we used meteorological fields simulated using the Advanced Research WRF model (WRF–ARW version 3.9.1.1; Skamarock et al., 2008). As meteorological initial and lateral boundary conditions for the WRF simulation, we used grid point value (GPV) data produced by the mesoscale forecast model (MSM) of the Japan Meteorological Agency (JMA) (MSM–GPV; JMA, 2019). The MSM–GPV data have 17 vertical layers, including the surface layer with a spatial resolution of $0.0625° \times 0.0500°$ and 16 pressure levels (from 1000 to 100 hPa) with a spatial resolution of $0.125° \times 0.100°$.

Although the MSM–GPV data provide 3-hourly forecast values, only the initial values of each forecasting cycle were used in this study. The initial and lateral boundary conditions of soil temperature and moisture were obtained from final operational global analysis and forecast data (GDAS/FNL) of the National Centers for Environmental Prediction (NCEP) with a spatial resolution of $0.25° \times 0.25°$ (NCEP, 2015). Because the spatial resolution of the MSM–GPV data at the 16 pressure levels was ~10 km, the WRF model was configured with two modeling domains (d01 and d02 in Fig. S1) with horizontal resolutions of

3 km (d01) and 1 km (d02), with one-way grid nesting. Domain 01 included not only the Kanto Plain but also the surrounding mountainous and marine areas, and domain 02 fully covered the TMA and was slightly larger than the domain used for the STILT simulations. We set 51 vertical levels extending from the surface up to 100 hPa. Land use information was taken from a dataset (veg_jstream) constructed by Japan's Study for Reference Air Quality Modeling (Chatani et al., 2018). To reduce computational effort, the WRF simulations were not run for the entire campaign period but for separate intervals of 2.5–5.5

consecutive days, which were determined so that they covered the EM27/SUN measurement days. Each simulation segment started at 12:00 UTC, and the first 12 h was considered spin-up time. Grid nudging toward the MSM–GPV data was applied to the wind field (uv), temperature (t), and the water vapor mixing ratio (q) at all levels in domain 01 with a nudging coefficient of $3.0 \times 10^{-4}$ s$^{-1}$ for each element. In domain 02, grid nudging of the wind field was applied at all levels with a nudging coefficient of $1.0 \times 10^{-4}$ s$^{-1}$, whereas nudging was applied to the temperature and water vapor mixing ratio only at the levels

above the simulated PBL with a nudging coefficient of $3.0 \times 10^{-5}$ s$^{-1}$. The simulations for domains 01 and 02 were carried out with integration time steps of 15 s and 5 s, with model outputs saved every 1 h and every 15 min, respectively. For use with the STILT model, the wind data for domain 02 were time-averaged over the output interval of the WRF model (Nehrkorn et al., 2010). Table 1 summarizes the model settings and physics options used for the reference inverse analysis.

   In previous studies using the WRF model, the physics options of the model were set according to the studied region and period

as well as the WRF version. In this study, we sought to select optimal physics options especially for the PBL scheme, the cumulus parameterization scheme, and the land surface model, all of which significantly impact the WRF calculation (Díaz-Isaac et al., 2018), by comparing WRF (and STILT) simulation results obtained with different physics options to measurement data (see Sect. 4.2 for the STILT simulation). The Kain–Fritsch cumulus parameterization scheme (Kain, 2004) was applied only in domain 01 (Table 1). Because cumulus parameterization is valid only for coarse grid resolutions (typically >10 km),

we also ran simulations without any cumulus parameterization scheme and found little difference in the simulation results obtained with and without a cumulus parameterization scheme. For the land surface model, we adopted the Rapid Update Cycle (RUC) model (Smirnova et al., 2016) because $XCO_2$ simulations using the RUC model reproduced the observations better than the other land surface models examined in Díaz-Isaac et al. (2018). We evaluated in detail the effect of different PBL schemes on the WRF simulation results because which PBL scheme is optimal depends on the location and season (e.g.,

Jeong et al., 2013). We compared the modeled wind fields with observational data from Automated Meteorological Data Acquisition System (AMeDAS) stations operated by the JMA (https://www.jma.go.jp/jma/en/Activities/amedas/amedas.html). Because wind fields directly influence atmospheric transport patterns, it is of particular importance to assess the model performance of the wind fields. Here, we compared wind speed and wind direction in WRF simulations among three different PBL schemes, the Mellor–Yamada–Janjić (MYJ) scheme (Janjić, 1994), the Mellor–Yamada Nakanishi Niino Level 2.5

(MYNN25) scheme (Nakanishi and Niino et al., 2009), and the Yonsei University scheme (Hong et al., 2006) with topographic



correction for surface winds (Jimenez and Dudhia, 2012) (YSU+topo), and we also compared the fifth-generation atmospheric reanalysis (ERA5) data at 0.25° spatial resolution (Hersbach et al., 2020) with data from the five AMeDAS stations within the TMA (Fig. 1): Saitama (35.8761° N, 139.5868° E, 18 m ASL), Tokyo (35.6916° N, 139.7532° E, 56 m ASL), Haneda (35.5636° N, 139.7896° E, 16 m ASL), Chiba (35.6028° N, 140.1040° E, 51 m ASL), and Kisarazu (35.3623° N, 139.9402°

E, 68 m ASL). Comparison of wind speed and wind direction between the models and observations (see Fig. 4 for the Tokyo site) showed that the model data could generally reproduce the observed temporal variations in the wind fields and demonstrated the model's capability to simulate reasonable meteorological fields for driving the transport of trace gases. Tables 2 and 3 summarize the mean differences (biases) between the models and observations and their standard deviations for wind speed and wind direction, respectively. The WRF MYJ scheme had the lowest bias in wind speed and the smallest standard

deviation in wind direction, whereas the ERA5 results were the best for the standard deviation in wind speed and the bias in wind direction. Among the tested PBL schemes of the WRF model, the wind fields obtained with the MYJ scheme were optimal. $XCO_2$ simulations using these meteorological fields are assessed in Sect. 4.2.

### 3.3 Anthropogenic and biogenic $CO_2$ fluxes

For the prior estimate of anthropogenic $CO_2$ emissions, we used the 2020b version of the Open-source Data Inventory for

Anthropogenic $CO_2$ (ODIAC 2020b), which is a global high-resolution (30 arcsec) monthly fossil fuel $CO_2$ emissions database (Oda and Maksyutov, 2015; Oda et al., 2018). The high-resolution ODIAC emission map was created by spatially disaggregating the national $CO_2$ emissions using the large point source database and proxy data associated with emissions. The locations and magnitudes of large point source emissions in ODIAC are taken from the Carbon Monitoring for Action (CARMA) database (https://www.cgdev.org/topics/carbon-monitoring-action), and the rest of the national emissions

(nonpoint source emissions) are distributed based on the spatial patterns of night lights data collected by satellites. We pinpointed large point sources such as power plants and steel plants on the $CO_2$ emission map of ODIAC 2020b in March 2016 with the aid of high-resolution satellite imagery (Fig. 5a). The locations of large point sources in the ODIAC are not exact, likely because of the large uncertainty of the $CO_2$ emission source information in the CARMA database (Gurney et al., 2019). We therefore customized the ODIAC data by a two-step process. First, grid cells with $CO_2$ emissions larger than a given

threshold ($>10^4$ tons of carbon per month) were replaced with the averaged value of the neighboring eight grid cells. We regarded these secondary emissions as nonpoint source emissions. Second, annual emissions from large point sources, which are available upon request from the Ministry of the Environment of Japan under the GHG Emissions Accounting, Reporting, and Disclosure System (https://ghg-santeikohyo.env.go.jp/), were converted to monthly values and added to the nonpoint source emissions. The emission map corrected for the large point sources (referred to as the LPS-corrected ODIAC; Fig. 5b)

was used as the prior estimate. Large point source and nonpoint source emissions comprised 37 % and 63 %, respectively, of the LPS-corrected ODIAC data. In addition to this correction, weekly and diurnal scaling factors from the Temporal Improvements for Modeling Emissions by Scaling (TIMES) model developed by Nassar et al. (2013) were applied to the ODIAC data to temporally downscale the monthly ODIAC product.

A bottom-up fossil fuel emission inventory in Japan with a spatial resolution of approximately 1 km × 1 km, the Multiscale

Overlap Scheme for Analyzing national Inventory of anthropogenic $CO_2$ (MOSAIC), was used in a complementary manner (described in Sect. 4.3) (Saito et al., 2023). This emission inventory provides monthly data obtained by using Japanese government statistics for all socio-economic activities of Japanese society. The inventory comprises fossil fuel $CO_2$ emissions from eight sectors: electricity generation, waste incineration, civil aviation, waterborne navigation, road transportation, industrial and commercial sources, residential sources, and agricultural machine use. The locations and emission magnitudes

of the large point sources in the electricity generation sector were corrected using the same method applied to the ODIAC data. Note that this study used MOSAIC emission data for 2015, which were the only MOSAIC data available when the study was carried out.



To account for the influence of biogenic $CO_2$ on the observed $\Delta XCO_2$ values, we used biogenic $CO_2$ fluxes calculated by a terrestrial biosphere model. Specifically, hourly net ecosystem exchange (NEE) data from the Vegetation Integrative SImulator for Trace gases (VISIT) model, referred to as VISITc, combined with green vegetation fraction (GVF) data, were adopted as the biogenic $CO_2$ flux data. The NEE data reflect the $CO_2$ flux between the terrestrial biosphere and the atmosphere and are obtained as the difference between ecosystem respiration (RE) and gross primary productivity (GPP) in the VISITc data (RE – GPP). Whereas the original VISIT products provided monthly fluxes with a 0.5° spatial resolution (Ito and Inatomi, 2012), the VISITc products provide hourly-resolved fluxes composited with meteorological input data from the Climate Forecast System Reanalysis (CFSR) versions 1 and 2 (Saha et al., 2010) and ERA-Interim (Dee et al., 2011). In addition, the CFSR version 1 product has a global Gaussian T382 grid; thus, VISITc has the same spatial resolution as the CFSR data (i.e., approximately 0.31° × 0.31°). The original VISIT model simulates the terrestrial biogeochemical cycle with a monthly resolution considering minor carbon flows, such as the effect of land-use change and fire disturbance, that directly affect variability in GPP and RE (Ito, 2019). These effects, however, were not adopted in VISITc, so this study scaled the GPP and RE data derived from VISITc to those of original VISIT in every month and every grid. Furthermore, to better characterize the spatial distribution of biogenic $CO_2$ fluxes, we spatially downscaled the hourly VISITc NEE data to the STILT footprint grids (i.e., approximately 1 km × 1 km) using GVF data with an approximately 4-km spatial resolution, which are produced daily from the previous 7 days of data from the Visible Infrared Imaging Radiometer Suite (VIIRS) sensor onboard the Suomi National Polar-orbiting Partnership satellite (Ding and Zhu, 2018). The GVF parameter, which represents how much downward solar insolation is intercepted by the canopy, is used as a scaling parameter for the biogenic flux. Following the method of Ye et al. (2020), the original GVF data (Fig. 6a) were first averaged into the VISITc grid. Then, the VISITc NEE data (Fig. 6b) and the re-gridded GVF data were interpolated bilinearly into the 1 km × 1 km grid. From the bilinearly interpolated NEE and GVF data and the original GVF data, the NEE data with the 1 km × 1 km resolution were created (Fig. 6c). Finally, the downscaled NEE data were scaled so that all NEE data from the TMA were preserved before and after the downscaling.

### 3.4 Inversion methodology

As described in Sect. 3.1, $\Delta XCO_2$ values measured at a given location are quantitatively related to the presumed surface $CO_2$ fluxes via the forward model $\boldsymbol{H}$, representing the atmospheric transport:

$$\boldsymbol{y} = \boldsymbol{H}(\boldsymbol{x}, \boldsymbol{b}) + \boldsymbol{\varepsilon}, \tag{1}$$

where $\boldsymbol{y}$ denotes the measurement vector ($n \times 1$), $\boldsymbol{x}$ is the state vector to be retrieved ($m \times 1$), $\boldsymbol{b}$ is the fixed state vector, and $\boldsymbol{\varepsilon}$ is the measurement error vector. The surface $CO_2$ fluxes can be estimated inversely from the observed $\Delta XCO_2$ values, together with their associated uncertainties, through an optimization procedure. In the present study, the state vector $\boldsymbol{x}$ includes spatially resolved nonpoint source emissions and a scaling factor of the large point source emissions. Because the geolocations of large point sources are precisely prescribed (at the grid cell scale of ~1 km), the emissions from the large point sources are treated separately from the nonpoint source emissions, which have large uncertainty in their spatial distribution. To simplify the inversion, the biogenic flux was allocated to the state vector $\boldsymbol{b}$, because the contribution of biogenic flux to the simulated $\Delta XCO_2$ was small (Sect. 4.2). For the nonpoint source emissions, we decided to optimize the logarithm of the emission flux, with the prior errors following a Gaussian distribution, because the emissions from each grid cell differ by a couple of orders of magnitude, and the optimization of emissions at linear scale might lead to unphysical negative posterior emissions. These anthropogenic $CO_2$ emissions (i.e., $\boldsymbol{x}$) were estimated based on a Bayesian framework by minimizing the cost function, which consists of two terms related to the measurements and prior emissions:

$$J(\boldsymbol{x}) = \left(\boldsymbol{y} - \boldsymbol{H}(\boldsymbol{x})\right)^T \mathbf{S}_\varepsilon^{-1}\left(\boldsymbol{y} - \boldsymbol{H}(\boldsymbol{x})\right) + (\boldsymbol{x} - \boldsymbol{x_a})^T \mathbf{S_a}^{-1}(\boldsymbol{x} - \boldsymbol{x_a}), \tag{2}$$



where $x_a$ is the prior vector of $x$; $S_\varepsilon$ is the model–observation mismatch covariance (or the measurement error) matrix ($n \times n$); and $S_a$ is the priori error covariance matrix ($m \times m$). Because the state vector $x$ includes the logarithm of nonpoint source emissions, the inverse problem is nonlinear, and an iterative approach was used to estimate the $CO_2$ emissions:

$$x_{l+1} = x_l + \left[ K^T S_\varepsilon^{-1} K + (1 + \gamma) S_a^{-1} \right]^{-1} \left[ K^T S_\varepsilon^{-1} (y - H(x_l)) - S_a^{-1} (x_l - x_a) \right], \tag{3}$$

where $K$ represents the Jacobian matrix ($n \times m$), which corresponds to a footprint obtained from the WRF–STILT model, $l$ is the iteration number, and $\gamma$ is the Levenberg–Marquart parameter (Rodgers, 2000). The posterior error covariance matrix was calculated with the following equation:

$$\hat{S} = (K^T S_\varepsilon^{-1} K + S_a^{-1})^{-1}. \tag{4}$$

The averaging kernel matrix represents the sensitivity of the posterior solution $\hat{x}$ to the true emission state:

$$A = \frac{\partial \hat{x}}{\partial x} = I - \hat{S} S_a^{-1}, \tag{5}$$

where $I$ is the identity matrix. The degree of freedom for signal (DOFS), which is the trace of the averaging kernel, indicates the number of independent pieces of information retrieved from the observing system. As a measure of the uncertainty reduction after the inverse analysis, the difference between the prior flux uncertainty and the posterior flux uncertainty relative to the prior flux uncertainty, $r$, can be used:

$$r = \left( 1 - \frac{\sqrt{\text{diag}[\hat{S}]}}{\sqrt{\text{diag}[S_a]}} \right) \times 100. \tag{6}$$

Because we applied weekly and diurnal correction factors to the prior ODIAC monthly emissions using the TIMES model, we optimized one static emission distribution during the campaign period, assuming that the temporal variation of the emissions followed the TIMES model. Considering the numbers of the observation sites and the $\Delta XCO_2$ data available for the inversion ($n = 654$), we aggregated the nonpoint source emissions and footprints in the original 1 km × 1 km ($0.0083° \times 0.0083°$) grids to $0.025° \times 0.025°$ grids ($m = 1921$). In the sensitivity test, the resolutions of nonpoint source emissions and footprints were further lowered to $0.05° \times 0.05°$ grids ($m = 481$) and $0.1° \times 0.1°$ grids ($m = 121$) (Sect. 4.4).

## 4 Results and discussion

### 4.1 $XCO_2$ measurements

During the 2016 Tokyo campaign, the daily minimum $XCO_2$ values observed by the four spectrometers increased gradually from 403 to 405 ppm as a result of seasonal variation, whereas the daily maximum values showed large day-to-day variation with peaks of up to 415 ppm (Fig. 3a). The $XCO_2$ values observed at Saitama and Sodegaura from 16 February to 6 April were generally higher than those observed at Tsukuba. To characterize the diurnal variation in $XCO_2$ at each observation site, we examined the diurnal variation in $XCO_2$ differences ($XCO_2^{\text{Diff}}$) from daily background values common to the three sites. The daily background $XCO_2$ values were assumed to be the 5 percentile values of the Tsukuba TCCON measurements throughout each day. For days when measurements at Tsukuba were not available, average CarbonTracker CT2019B $XCO_2$ data (Jacobson et al., 2020) of two grids centered over the ocean east of the TMA (35.0° N, 142.5° E and 37.0° N, 142.5° E) were used as the background values. We note that from February to April 2016 (average of 12:00–15:00 LT), the mean difference between the CarbonTracker data (13:30 LT) and the Tsukuba TCCON data (CarbonTracker minus TCCON) was −0.18 ppm with a standard deviation of 0.72 ppm. The maximum $XCO_2^{\text{Diff}}$ value was 9.5 ppm at Saitama and 9.3 ppm at Sodegaura. The average $XCO_2^{\text{Diff}}$ value per 15-min bin was calculated for each site using the entire field campaign dataset (Fig. 7). The $XCO_2^{\text{Diff}}$ values at



Tsukuba gradually decreased over time with small standard deviations. This finding reflects the absence of other large emission sources around Tsukuba and the moderate effect of photosynthesis. The measurements at Saitama and Sodegaura showed larger $XCO_2^{Diff}$ values than those at Tsukuba, and $XCO_2^{Diff}$ values increased over time from approximately 08:00 JST. We note that the high early morning values at Saitama may reflect an airmass-dependent bias. Although the $XCO_2^{Diff}$ variability at Sodegaura was smaller than that at Saitama, some data bins with large standard deviations likely represent occasional

influences of emissions from nearby large point sources.

The daily minimum $XCH_4$ values (Fig. 3b) showed relatively larger temporal fluctuation than those of $XCO_2$ because of synoptic-scale events (e.g., 22–28 February). Although detailed analysis of $XCH_4$ data is beyond the scope of this study, in the case study evaluating the WRF–STILT simulation presented in the next section, the $XCH_4$ values are used.

**4.2 XCO₂ simulations**

As described in Sect. 3.1, the $XCO_2$ enhancement ($\Delta XCO_2$) was calculated from the column-averaged footprint and the surface fluxes from nonpoint sources, large point sources, and biological activity. For the large point source emissions and biogenic flux, the $\Delta XCO_2$ values were calculated from footprints with a spatial resolution of approximately 1 km × 1 km (0.0083° × 0.0083°). For nonpoint source emissions, however, we re-gridded the original footprints to a spatial resolution of 0.025° × 0.025°, along with the nonpoint source flux data, to degrade the spatial resolution for the inverse analysis. The simulations

were performed when EM27/SUN measurements were available for more than 2 h per day and when two or more of the 15-min averaged data showed $XCO_2$ differences of at least 1 ppm between the urban (i.e., Saitama or Sodegaura) and Tsukuba TCCON site. At Saitama and Sodegaura, these conditions were satisfied on 15 and 11 days, respectively, between 16 February and 23 March 2016. In addition, the simulations were limited to the time period from 09:00 to 17:00 JST, when synchronous measurements at the three sites were made and the airmass-dependent variation in $XCO_2$ was moderate. Because the time

period when the forward simulations and the subsequent inverse analysis were performed was approximately one month during February and March 2016, for the prior of the anthropogenic emissions we used the average of the ODIAC data in February and March 2016. Figure S2 in the Supplement shows the $\Delta XCO_2$ values at the three sites separately simulated using nonpoint sources, large point sources, and biogenic fluxes. We calculated the contributions of the respective fluxes to the simulated $\Delta XCO_2$ at each site (Table 4). The contribution from nonpoint source emissions dominated the simulated $\Delta XCO_2$ for the

Saitama and Tsukuba sites. Because the Sodegaura site is located near large point sources, the closest one being approximately 4 km away, the contribution from large point sources was larger at Sodegaura than at Saitama. Because our observations were made from late winter to early spring, the biogenic flux contribution was relatively small, but not negligible, especially at the Saitama site.

To compare the observed $XCO_2$ data with the simulations, we must add background $XCO_2$ values to the anthropogenic and

biogenic $\Delta XCO_2$ values simulated by WRF–STILT. Because the $XCO_2$ values observed at Tsukuba were systematically lower than those at the other urban sites, as can be seen from the $XCO_2^{Diff}$ values in Fig. 7, which were calculated from the background common to the three sites, the $XCO_2$ values at Tsukuba were assumed to approximately represent background air. However, the observations at Tsukuba are not strictly background; they are affected by emissions from the TMA. We therefore obtained the background $XCO_2$ values by subtracting the simulated $\Delta XCO_2$ values at the Tsukuba site from the observed $XCO_2$ values

at the Tsukuba site. Additionally, the background values were presumed to be common to the three sites, given the proximity of the sites. When there were no data available from the Tsukuba site, CarbonTracker CT2019B $XCO_2$ data were used for background values independent of local time (see Sect. 4.1).

We compared $XCO_2$ data between the WRF–STILT forward simulations and the EM27/SUN observations at Saitama and Sodegaura (Figs. 8 and 9). Here, the EM27/SUN data were averaged in 15-min bins, whereas the simulated data correspond

to the sum of the WRF–STILT $\Delta XCO_2$ value every 15 min at each site and the background $XCO_2$ value. In most cases, the





forward simulations captured well the observed temporal variation of XCO₂. However, in some cases they failed to reproduce the diurnal variations. As shown in Fig. 9d, we found it difficult to correctly capture the timing of short-term (<1 h) XCO₂ enhancements, which were likely caused by the plume from the large point sources near the Sodegaura site. Similarly, the WRF–STILT simulation for Saitama on 3 March 2016 was not able to capture the XCO₂ enhancement in the late afternoon
(Fig. S3a). STILT simulations conducted using ERA5 data and WRF data with different PBL schemes (not shown) showed similar tendencies. Furthermore, even when we changed the emission data from ODIAC to MOSAIC, the discrepancy was not reduced. In addition, we investigated the XCH₄ data, for which a diurnal variation similar to XCO₂ was observed. A WRF–STILT simulation using the Emissions Database for Global Atmospheric Research (EDGAR) version 6 (Crippa et al., 2019) as the CH₄ emission inventory also could not capture the XCH₄ enhancements in the late afternoon (Fig. S3b). Therefore, we
attribute this large model–observation discrepancy to errors in the WRF-STILT model rather than to the emission data. These additional simulations indicate that further improvement of the WRF simulation (i.e., assimilation of measurement data) is necessary for more accurate generation of meteorological fields. In our inverse analysis, this large modeling error was considered when setting the measurement error covariance matrix, as described in the next section.

## 4.3 Inversion settings

The measurement vector **y** consists of the differences between XCO₂ values measured at the urban sites (Saitama or Sodegaura, $XCO_2{}^{Urban}_{EM27/SUN}$) and the Tsukuba site ($XCO_2{}^{Tsukuba}_{TCCON}$). Provided that XCO₂ values at each site can be represented by the sum of ΔXCO₂ values ($\Delta XCO_2{}^{Urban}_{STILT}$ or $\Delta XCO_2{}^{Tsukuba}_{STILT}$) and a background value common to the three sites ($XCO_2{}^{BG}$), the difference in XCO₂ values between each urban site and the Tsukuba site is equal to the difference in their ΔXCO₂ values:

$$XCO_2{}^{Urban}_{EM27/SUN} - XCO_2{}^{Tsukuba}_{TCCON} = \Delta XCO_2{}^{Urban}_{STILT} + XCO_2{}^{BG} - \left( \Delta XCO_2{}^{Tsukuba}_{STILT} + XCO_2{}^{BG} \right)$$

$$= \Delta XCO_2{}^{Urban}_{STILT} - \Delta XCO_2{}^{Tsukuba}_{STILT}. \tag{7}$$

Consequently, to simulate the XCO₂ difference between each pair of sites corresponding to the measurement vector **y**, we should calculate the ΔXCO₂ values at the two sites and take their difference. Figure 10 shows the mean ΔXCO₂ contribution from nonpoint source emissions during the field campaign, which was calculated from the absolute value of the ΔXCO₂ difference, expressed by the right-hand side of Eq. 7. We limit the domain for estimating nonpoint source emissions to the
urban domain indicated by the magenta rectangle in Fig. 10 (hereinafter referred to as the inversion domain), where the contributions of each grid cell to the modeled ΔXCO₂ are relatively large.

To construct the prior error covariance matrix **Sₐ**, we compared the ODIAC emission data used as the prior estimate (Fig. 11a) with the MOSAIC emission data (Saito et al., 2023). Although the two databases have similar spatial resolution, it is not the same, so we re-gridded the MOSAIC emission data into the ODIAC grid (Fig. 11b). Then, we aggregated both datasets into
the 0.025° × 0.025° grid used for the inverse analysis. Figure 11c shows the difference between the aggregated datasets, calculated as (ODIAC – MOSAIC) / (0.5 × (ODIAC + MOSAIC)) × 100. The difference between the spatial distributions of the ODIAC and MOSAIC data is based solely on nonpoint sources because the large point sources are common to the two datasets. Similar large spatial differences also exist among other emission inventories (Gately and Hutyra, 2017) because this kind of emissions database uses geospatial information or physical proxies to allocate the spatial distributions of emissions.
Considering the standard deviation of the difference between the two datasets, we set the diagonal elements of **Sₐ** to 85 % of the prior emission values. For the scaling factor of the large point source emissions, we set the uncertainty to 15 % based on the temporal variability of monthly liquid natural gas consumed by natural gas–fired power plants of the Tokyo Electric Power Company         Holdings,         which         were         available         up         to         March         2016 (https://www.enecho.meti.go.jp/statistics/electric_power/ep002/results_archive.html).
The off-diagonal elements of the prior error covariance matrix, which represent the spatial coherence between the prior flux



uncertainties in different grid cells, were calculated according to a model of exponential decay with distance between grid cells (e.g., Lauvaux et al., 2016; Lopez-Coto et al., 2020). Thus, the element $[i, j]$ of the prior error covariance matrix was given as,

$$\mathbf{S_a}[i, j] = \sigma_i\, \sigma_j \exp\left(- d_{i,j}/L_{\mathrm{s}}\right), \tag{8}$$

where $\sigma_i$ ($\sigma_j$) represents the uncertainty of the emissions in grid cell $i$ ($j$), $d_{i,j}$ is the distance between grid cells $i$ and $j$, and $L_{\mathrm{s}}$ is

the spatial correlation length of the prior flux uncertainties. To determine the spatial correlation length of the prior flux uncertainties, we computed semi-variograms of the differences in nonpoint source elements between the two emissions datasets in the inversion domain, and then we fitted an exponential model to the semi-variograms with the distance between grid cell pairs limited to 30 km (Mallia et al., 2020). This analysis yielded a correlation length of approximately 10 km (Figure S4), which is equivalent to that in New York (Pitt et al., 2022) and Salt Lake City (Mallia et al., 2020).

To estimate the measurement error (or model–observation mismatch) covariance matrix $\mathbf{S_\epsilon}$, we used the residual error method of Heald et al. (2004). In this method, the residual errors between the $XCO_2$ values measured by EM27/SUN spectrometer and those simulated by WRF–STILT using the prior data were computed, and the variance of the residual over the campaign period was used to represent the diagonal elements of $\mathbf{S_\epsilon}$. Figure S5 is a scatter plot between the measured and simulated $XCO_2$ values over the campaign period; here, the standard deviation of the residual, $\sigma_\epsilon$, is 1.31 ppm. As described in Sect. 4.2, the large

model–observation discrepancies stem from the difficulty of simulating the $CO_2$ plumes emitted from large point sources and local meteorological conditions. When the residual between the simulation and observation was more than three times $\sigma_\epsilon$, the measurements were screened out by greatly increasing the uncertainty. An exponential covariance model in time was selected with a temporal correlation length of 1 h based on the value reported for continuous $CO_2$ observations in urban areas (Turner et al., 2020).

Although the residual error method provides a realistic model–observation mismatch, we also estimated individual uncertainties in our model–observation system, consisting of uncertainties in the measurement data, transport modeling, biogenic flux, and background value. We assumed the uncertainty in measurement data to be the standard deviation of the differences between the EM27/SUN $XCO_2$ data acquired by side-by-side instruments (Sect. 2). The standard deviation of the bias-corrected $XCO_2$ differences between the SN38 and SN44 EM27/SUN spectrometers was 0.16 ppm. To estimate the

uncertainty in $XCO_2$ due to the transport modeling error, we ran $XCO_2$ simulations using the WRF data with different PBL schemes (see Sect. 3.2) and the ERA5 data. The mean biases and the standard deviations of the difference between the EM27/SUN measurements and the STILT simulations from the prior fluxes are listed in Table 5 (Prior $XCO_2$ difference). Whereas there was no large difference in the standard deviation among the three simulations using the WRF data (1.31–1.40 ppm), the standard deviation for the simulation using ERA5 was 2.74 ppm, more than 1 ppm larger than that of the simulations

using WRF. This large value was because the $\Delta XCO_2$ values at Sodegaura on 23 March 2016 simulated from the ERA5 data showed a rather large peak (~20 ppm) caused by incidental contamination from the nearby large point sources that was not present in the actual measurement data. When the data for that site and day were excluded, the standard deviation decreased to 1.85 ppm. The $XCO_2$ uncertainty resulting from transport modeling, estimated as the standard deviation of the differences between the $XCO_2$ values simulated using the WRF and ERA5 meteorological fields, was 1.65 ppm. To estimate the

uncertainty in $XCO_2$ resulting from the biogenic flux error, we calculated $XCO_2$ values over the campaign period for four types of biogenic fluxes (VISITc and three others) with differing spatial and temporal resolutions (Table S1) but with other input parameters unchanged. Simple Biosphere Model version 4.2 (SiB4, Haynes et al., 2021) and Biosphere model integrating Eco-physiological And Mechanistic approaches using Satellite data (BEAMS, Sasai et al., 2005) are both terrestrial biosphere models, whereas CarbonTracker version CT2019B (Jacobson et al., 2020) is from a data assimilation system in which the

biogenic and oceanic fluxes are optimized. The average standard deviation across the $XCO_2$ values calculated using the four biogenic fluxes, 0.09 ppm, was regarded as the $XCO_2$ uncertainty resulting from the biogenic flux. When Tsukuba TCCON data were not available and CarbonTracker data were used instead, the uncertainty in the background value rose. We assumed



that the XCO$_2$ uncertainty resulting from the background value was represented by the standard deviation of the XCO$_2$ difference between the Tsukuba TCCON data and the CarbonTracker data and estimated the uncertainty as 0.72 ppm (see Sect.

4.1). These evaluations revealed that the uncertainty in transport modeling was dominant, followed by the uncertainty in background value.

### 4.4 Posterior fluxes

Figure 12a shows the posterior nonpoint source CO$_2$ emissions, estimated using the settings for the reference inversion (i.e., case #0 in Table 5 using the ODIAC as the prior data and meteorological fields from the WRF model with the MYJ PBL

scheme for footprint calculations). The total DOFS from the reference inversion was 6.49, of which 5.73 is for spatially resolved emissions and 0.76 is for the large point source emissions. The spatial pattern of the optimized emissions still largely resembled the prior estimate pattern (Fig. 12). In large parts of Tokyo and Kanagawa, the emissions were revised downward, whereas in Saitama, Ibaraki, and northern Chiba, the emissions became larger. Because the mean bias in XCO$_2$ values simulated from the prior emission flux was originally small, the emissions from the central TMA region were adjusted

downward, and the emissions from the other regions were adjusted upward. The spatial distribution of the changes from the prior flux was partly in agreement with the spatial differences between the MOSAIC and ODIAC emission data (Fig. 11c). Because the locations of large point sources were corrected in the prior emissions (Sect. 3.3), the difference in the spatial distribution between the prior and posterior emissions may be due to the unrepresentativeness of the spatial proxy (i.e., night lights) used in the ODIAC data. As an indication of the efficiency of the inversion, we evaluated to what extent the differences

between the XCO$_2$ simulations and observations were improved by using the posterior fluxes. The XCO$_2$ values simulated from the posterior fluxes were in better agreement with the observations than those simulated from the prior fluxes (Fig. 9). The mean bias in XCO$_2$ simulations against observations decreased from 0.30 to –0.03 ppm, but the RMSE decreased only slightly, from 1.31 ppm to 1.21 ppm (Fig. S5). This slight RMSE reduction is because the emission distribution was estimated on a monthly basis, whereas the individual model–observation discrepancies were governed by the transport modeling error.

Next, we compared the estimated total emissions in the TMA with the emission inventories. The total emissions correspond to the domain-aggregated emission flux during the campaign period (i.e., from February to March 2016). Figure 13 shows the total emissions calculated from the prior flux and the posterior flux in the reference inversion. The error bars (uncertainties at the 95 % confidence level) of prior and posterior total emissions are based on the respective error covariance matrices and were obtained by summing the emission uncertainties in each grid cell and the uncertainty of the large point source emission

in quadrature. The posterior large point source emissions were adjusted downward by 14.4 % compared with the prior emissions (i.e., scaling factor of 0.856), and the posterior nonpoint source emissions were adjusted upward by 10.4 %. Consequently, the difference between the prior and posterior total emissions was approximately 1 %. Although the change in the total emissions was relatively small, the inversion led to a reduction of the uncertainty in the total emissions by a factor of ~2 (i.e., the uncertainty at the 95 % confidence level decreased from 11.3 % to 5.2 %).

We present here the results of emission estimates obtained for cases with different inversion settings (Table 5): case #1, large point source emissions fixed; cases #2a–c, footprints calculated from different meteorological fields used; cases #3a and #3b, prior uncertainty halved or doubled; cases #4a and #4b, spatial correlation length changed; case #5, EDGAR version 6 (0.1° × 0.1° spatial resolution) without large point source correction used as the prior estimate; and cases #6a and #6b, spatial resolution of the inversion domain coarsened to 0.05° or 0.1° (i.e., 2 or 4 times the reference case). For the case #5 and #6

inversions, the prior uncertainty and the spatial correlation length were re-determined as described above. The total emissions, scaling factor of large point source emissions, and ΔXCO$_2$ bias between the simulation and observations and its standard deviation for each case are summarized in Table 5. For case #1, the inversion in which the large point source emissions were fixed, both the mean bias and the standard deviation of the posterior XCO$_2$ simulations against observations were equivalent to those of the reference inversion (case #0). Although the scaling factor of large point source emissions for the reference



inversion was 0.856, total emissions in case #1 were 5.3 % larger than those in case #0. The posterior $XCO_2$ simulation results obtained with different meteorological fields (cases #2a–c) indicated that the biases and standard deviations were improved compared to the prior $XCO_2$ simulations, irrespective of the meteorological field. Among them, use of the WRF model with the MYJ scheme resulted in the smallest standard deviations for not only the prior but also the posterior $XCO_2$ simulations. When the prior uncertainty (cases #3a and #3b) and its correlation length (cases #4a and #4b) were changed, the mean biases

and the standard deviations of the posterior $XCO_2$ simulations were comparable to the reference inversion. However, the prior uncertainty had a larger impact on the total emission estimates than the spatial correlation length. The inversion using EDGAR as the prior emission inventory (case #5) resulted in a posterior $XCO_2$ simulation with a low bias of 0.16 ppm and a standard deviation of 1.27 ppm; this simulation underestimated the total emissions by 15.8 % compared with the reference inversion. This result implies that the use of emission data with a low spatial resolution introduces additional uncertainty into $XCO_2$

modeling (Fig. S6). Similarly, reducing the spatial resolution of the ODIAC data slightly (cases #6a and #6b) degraded both the mean bias and the standard deviation of the posterior $XCO_2$ simulations. In the case of the reference inversion, the number of measurement data points was considerably smaller than the number of grid cells whose emissions were optimized. Although the number of the grid cells with a spatial resolution of 0.05° and 0.1° was equivalent to or lower than the number of measurement data points, respectively, the total DOFS slightly decreased (to 5.84 for 0.05° and 5.05 for 0.1°) because the prior

uncertainty and the spatial correlation length also changed. The posterior total emissions did not differ greatly from those of the reference inversion. Figure 13 shows the ensemble mean of the total emissions and its uncertainty at the 95 % confidence level, estimated from the scatter of these inversion results. For comparison, the total emissions from the original ODIAC data and the original and LPS-corrected MOSAIC data are also displayed. The ensemble mean total emissions are in agreement with the original and LPS-corrected MOSAIC emissions within the uncertainty of the ensemble inversions.

We compared our results with those of a previous $CO_2$ inversion study for the TMA (Pisso et al., 2019; Babenhauserheide et al. 2020) and with annual emissions in fiscal year (FY) 2015 (April 2015 to March 2016) reported by each administrative division in the TMA. Here, we calculated the total $CO_2$ emissions in the Tokyo Metropolis by integrating emissions in the grid cells within its administrative boundaries. Additionally, because our inversion domain included almost the whole area of the Tokyo Metropolis and of Kanagawa, Chiba, and Saitama Prefectures, the total emissions from these four administrative

divisions (referred to as southern Kanto) were also calculated. The total emissions estimated by our reference inversion were 56.6 Mt-$CO_2$ yr$^{-1}$ for the Tokyo Metropolis and 277.8 Mt-$CO_2$ yr$^{-1}$ for southern Kanto (Fig. S7), and these emissions are smaller by 29 % and 50 %, respectively, than those estimated by Pisso et al. (2019). We note that although Pisso et al. (2019) estimated mean emissions for 2005–2009, the difference between the FY2015 emissions and the FY2005–2009 mean emissions reported by the Tokyo Metropolis is less than 1 %

(https://www.kankyo.metro.tokyo.lg.jp/en/climate/index.files/Tokyo_GHG_2019.pdf). Babenhauserheide et al. (2020) estimated $CO_2$ emission of 256 ± 77 Mt-$CO_2$ yr$^{-1}$ for the urban area around Tokyo. Our emission estimate for southern Kanto was in reasonable agreement with the result of Babenhauserheide et al. (2020), although the comparison is not exact because of the discrepancy in the areas where the $CO_2$ emissions were calculated. The total emissions in FY2015 reported from each administrative division were 60.3 Mt-$CO_2$ yr$^{-1}$ for the Tokyo Metropolis and 250.6 Mt-$CO_2$ yr$^{-1}$ for southern Kanto (Table

S2); these values show remarkable agreement with our posterior estimates from the reference inversion. Furthermore, our posterior estimate for the Tokyo Metropolis lies between the ODIAC and MOSAIC inventory data. The relationship between our posterior estimate and the inventory data for southern Kanto is similar to that for the TMA shown in Fig. 13, because southern Kanto includes most of the TMA as defined in this study. Thus, these comparisons demonstrate that our top-down approach was able to properly constrain $CO_2$ emissions in this urban area.

**5 Conclusion**



We conducted a field campaign to estimate $CO_2$ emissions in the TMA from February to April 2016 with two EM27/SUN spectrometers deployed at sites in Saitama and Sodegaura and the Tsukuba TCCON spectrometer. The $XCO_2$ values at Saitama and Sodegaura exhibited large enhancements compared with those at Tsukuba, and the mean diurnal variation of the enhancements showed a tendency to increase toward evening. The Lagrangian transport model STILT, which was driven by

WRF meteorological fields generated at a horizontal resolution of ~1 km, was used for simulating the $XCO_2$ enhancements resulting from anthropogenic (nonpoint source and large point source) emissions and biogenic fluxes. As the prior fluxes, the anthropogenic emissions from the ODIAC dataset were corrected by replacing the locations and emission magnitudes of large point sources with inventory data, whereas the biogenic flux from VISITc was downscaled using GVF data. We found that, for the TMA, the WRF model with the MYJ PBL scheme and the RUC land surface model yielded optimal results with regard

to both wind fields and the $XCO_2$ simulations. The $XCO_2$ forward simulation results using the prior fluxes highlight several factors that should be considered when designing an observation campaign or an operational network for ground-based column measurements for estimating urban emissions. Although the $XCO_2$ forward simulations generally showed good agreement with the observations, the comparison between the simulations and observations demonstrated some limitations in the modeling capability. As described in Sect. 4.2, in some cases, the simulations failed to reproduce the diurnal variation and to

capture the plume from nearby large point sources, primarily because of the transport modeling error, which included uncertainties in the meteorological field simulations (Figs. 9d and S3). Assimilating meteorological measurement data into the WRF calculation would be one way to reduce the modeling error. In a previous study, we conducted simultaneous measurements of $XCO_2$ and wind data with the EM27/SUN instruments and a Doppler lidar, respectively, co-located close to a thermal power plant in Japan (Ohyama et al., 2021). However, because not even the simulation using the measured wind

data and a simple dispersion model could reproduce the timing of the observed $XCO_2$ enhancement, we decided to adjust the wind directions as part of the optimization of emission fluxes. Thus, it is a great challenge to simulate the plumes from large point sources. At the Sodegaura site, where there are two large point sources within 10 km and two more within 15 km, the contribution from the large point sources in the TMA to the simulated $\Delta XCO_2$ is equivalent to the contribution from nonpoint sources (Table 4 and Fig. S2). These findings suggest that, for the purpose of estimating emissions from the entire city, the

locations of the EM27/SUN instruments should be selected to avoid proximity to large point sources or, through consideration of the dominant wind direction, to minimize the influences from large point sources.

Using these observational and modeling approaches along with their uncertainties, we developed an urban area-scale inversion system to estimate spatially resolved $CO_2$ emission at >3 km resolution and a suitable scaling factor for large point source emissions. The posterior $CO_2$ flux reduced both the mean bias and the standard deviation of the differences between the $XCO_2$

simulations and observations. Whereas the posterior total $CO_2$ emissions in the TMA from the reference inversion were consistent with those from the prior estimate with ~1 %, the posterior uncertainty was halved compared with the prior uncertainty. The ensemble mean of the posterior total $CO_2$ emissions agreed with the LPS-corrected ODIAC (prior) and MOSAIC data within the posterior uncertainty at the 95 % confidence level estimated from the ensemble scatter. We conclude that the EM27/SUN data could constrain urban-level $CO_2$ emissions and partially resolve the spatial distribution at monthly

scale. Because few EM27/SUN instruments were available for the 2016 Tokyo campaign, we deployed only two EM27/SUN instruments with consideration of the prevailing wind direction. The actual wind direction varied more than expected, with the result that about one month of data showed a wide range of sensitivity, as shown in Fig. 10. The deployment of additional instruments would enable more frequent (i.e., bimonthly or weekly) emission estimates. We plan to construct operational observation sites with EM27/SUN spectrometers in central Tokyo and the TMA suburbs. These data not only will help

operational estimation of $CO_2$ emissions in the TMA, thereby helping to verify emission reduction efforts, but also will validate GHG data from future satellite missions with small footprints and a wide swath width, such as Japan's GOSAT-GW (Global Observing SATellite for Greenhouse gases and Water cycle; https://gosat-gw.nies.go.jp/en/) and ESA's CO2M (Copernicus Anthropogenic Carbon Dioxide Monitoring mission; https://www.esa.int/ESA_Multimedia/Images/2022/03/CO2M).



**Author contributions.** MMF, IM, KS, TB, and FH designed the observation campaign. MMF, KS, and IM performed the
EM27/SUN measurements with the support of MW. IM operated the Tsukuba TCCON FTS. HO designed the inversion
framework and performed the analysis. MN and HO performed the WRF model simulation and processed the WRF data, TM
and MS produced the VISITc data, and HY and MS provided the MOSAIC data. HO, MMF, and IM contributed to scientific
discussion on the results of the analysis. HO prepared the manuscript and all authors reviewed the manuscript.

**Competing interests.** The contact author has declared that none of the authors has any competing interests.

**Acknowledgments.** We thank T. Nakatsuru for his cooperation in operating the EM27/SUN spectrometer at Sodegaura. The
Sodegaura City Hall cooperated in the installation of the observation equipment and in the data acquisition. We thank the KIT
Graduate School for Climate and Environment (GRACE) for supporting this analysis. The services of the COCCON central
facility were used for instrument quality control and calibration before the EM27/SUN spectrometers were delivered to the
operators. The simulations with the WRF and STILT models were performed using the supercomputer system of the NIES.
The MSM–GPV data were collected and distributed by the Research Institute for Sustainable Humanosphere of Kyoto
University (http://database.rish.kyoto-u.ac.jp). The AMeDAS data were obtained from the JMA. The ERA5 reanalysis product
were retrieved from the Copernicus Climate Change Service Climate Data Store (https://cds.climate.copernicus.eu), and they
were converted to NOAA's Air Resource Laboratory data format using the HYSPLIT utility era52arl
(https://www.ready.noaa.gov/HYSPLIT_data2arl.php). The CarbonTracker CT2019B results were provided by NOAA ESRL,
Boulder, Colorado, USA (http://carbontracker.noaa.gov). The BEAMS data were provided by Dr. K. Murakami of NIES. The
green vegetation fraction data were obtained from NOAA CLASS (https://www.avl.class.noaa.gov).




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



**Table 1.** Physics and model options of the WRF data used in the reference inversion.

| | |
|---|---|
| Model version | V3.9.1.1 (WRF Preprocessing System: V3.9.1) |
| Meteorological input data | JMA mesoscale model (MSM–GPV) data |
| (Initial and boundary conditions) | Soil: NCEP-FNL data |
| Land use information | veg_jstream (Chatani et al., 2018) |
| Model grid size | d01, 3 km; d02, 1 km; 51 vertical layers |
| Grid nudging | d01: whole layer for uv, t, q (see main text) |
| | d02: whole layer for uv; above PBL for t, q |
| Planetary boundary layer | Mellor–Yamada–Janjić (MYJ) scheme (Janjić, 1994) |
| Land surface model | Rapid Update Cycle (RUC) model (Smirnova et al., 2016) |
| Surface layer | Revised MM5 scheme (Jiménez et al., 2012) |
| Microphysics | Thompson scheme (Thompson et al., 2008) |
| Cumulus parameterization | Kain–Fritsch scheme (only d01) (Kain, 2004) |
| Shortwave | RRTMG scheme (Iacono et al., 2008) |
| Longwave | RRTMG scheme (Iacono et al., 2008) |

**Table 2.** Mean differences and their standard deviations ($1\sigma$) in wind speed (m/s) between model data (three WRF simulations and ERA5 reanalysis data) and the observational data at five AMeDAS sites (model minus AMeDAS). Bold letters indicate the best-case results among the models.

| Site | WRF/MYJ | WRF/MYNN25 | WRF/YSU+topo | ERA5 |
|---|---|---|---|---|
| Saitama | **0.16** ± 1.30 | 0.20 ± 1.29 | 0.43 ± 1.36 | 0.17 ± **1.22** |
| Tokyo | **0.22** ± 1.53 | 0.70 ± 1.96 | 0.39 ± **1.38** | 0.40 ± 1.42 |
| Haneda | −0.96 ± 1.75 | **−0.78** ± 1.89 | −0.84 ± 1.80 | −0.98 ± **1.67** |
| Chiba | **0.18** ± 1.64 | 0.53 ± 1.89 | −0.48 ± 1.66 | −0.20 ± **1.46** |
| Kisarazu | 1.00 ± **1.38** | 1.30 ± 1.69 | 1.20 ± 1.47 | **0.84** ± 1.41 |

**Table 3.** Mean differences and their standard deviations ($1\sigma$) in wind direction (degrees) between model data (three WRF simulations and ERA5 reanalysis data) and the observational data at five AMeDAS sites (model minus AMeDAS). Bold letters indicate the best-case results among the models.

| Site | WRF/MYJ | WRF/MYNN25 | WRF/YSU+topo | ERA5 |
|---|---|---|---|---|
| Saitama | −9.5 ± 64.9 | −14.0 ± 63.1 | −12.9 ± 65.2 | **2.3** ± **60.4** |
| Tokyo | **−0.6** ± 52.8 | −2.5 ± 55.1 | 1.5 ± **50.6** | 10.0 ± 52.4 |
| Haneda | −0.7 ± **52.0** | −4.1 ± 55.0 | **0.6** ± 56.1 | 14.7 ± 56.9 |
| Chiba | −6.3 ± **48.6** | −5.1 ± 49.9 | −3.7 ± 49.9 | **−1.9** ± 52.0 |
| Kisarazu | −13.1 ± **47.1** | −10.4 ± 49.1 | −12.7 ± 49.4 | **−9.7** ± 55.7 |



**Table 4.** Mean fractions of $\Delta XCO_2$ simulated using the three $CO_2$ fluxes ($\Delta XCO_2$ $^{NPS}$ for nonpoint source emission, $\Delta XCO_2$ $^{LPS}$ for large point source emission, and $\Delta XCO_2$ $^{Bio}$ for the biogenic flux) to the sum of $\Delta XCO_2$ $^{NPS}$, $\Delta XCO_2$ $^{LPS}$, and the absolute

value of $\Delta XCO_2$ $^{Bio}$ for each site.

| Site | $\Delta XCO_2$ $^{NPS}$ (%) | $\Delta XCO_2$ $^{LPS}$ (%) | $\Delta XCO_2$ $^{Bio}$ (%) |
|---|---|---|---|
| Tsukuba | 77.8 | 15.7 | 6.6 |
| Saitama | 83.6 | 9.2 | 7.3 |
| Sodegaura | 47.7 | 47.6 | 4.6 |

**Table 5.** Total $CO_2$ emissions from the TMA, scaling factors of large point source (LPS) emissions, and prior and posterior $XCO_2$ differences between the simulations and observations for the different meteorological data, prior emission data, prior

uncertainty ($\sigma_e$) and its spatial correlation length ($l_s$), and spatial resolution of the inversion domain ($r_s$).

| Case | Meteorological data + prior emission data | $\sigma_e$ (%) | $l_s$ (km) | $r_s$ (°) | Prior XCO₂ difference (ppm) | Posterior XCO₂ difference (ppm) | Total CO₂ emission (Mt-CO₂ d⁻¹) | Scaling factor of LPS emissions |
|---|---|---|---|---|---|---|---|---|
| #0 | WRF/MYJ + ODIAC | 85 | 10 | 0.025 | $0.30 \pm 1.31$ | $-0.03 \pm 1.21$ | 1.037 | 0.856 |
| #1 | WRF/MYJ + ODIAC (LPS fixed) | 85 | 10 | 0.025 | $0.30 \pm 1.31$ | $0.00 \pm 1.23$ | 1.092 | 1 (Fixed) |
| #2a | WRF/MYNN25 + ODIAC | 85 | 10 | 0.025 | $0.26 \pm 1.39$ | $-0.02 \pm 1.29$ | 0.990 | 0.820 |
| #2b | WRF/YSU+topo + ODIAC | 85 | 10 | 0.025 | $0.16 \pm 1.40$ | $-0.10 \pm 1.31$ | 1.014 | 0.830 |
| #2c | ERA5 + ODIAC* | 85 | 10 | 0.025 | $-0.31 \pm 1.85$ | $-0.25 \pm 1.46$ | 0.846 | 0.537 |
| #3a | WRF/MYJ + ODIAC | 50 | 10 | 0.025 | $0.30 \pm 1.31$ | $0.00 \pm 1.23$ | 0.954 | 0.817 |
| #3b | WRF/MYJ + ODIAC | 120 | 10 | 0.025 | $0.30 \pm 1.31$ | $-0.04 \pm 1.20$ | 1.118 | 0.863 |
| #4a | WRF/MYJ + ODIAC | 85 | 5 | 0.025 | $0.30 \pm 1.31$ | $0.01 \pm 1.21$ | 1.045 | 0.818 |
| #4b | WRF/MYJ + ODIAC | 85 | 20 | 0.025 | $0.30 \pm 1.31$ | $-0.06 \pm 1.22$ | 1.013 | 0.862 |
| #5 | WRF/MYJ + EDGAR | 95 | 14 | 0.025 | $0.06 \pm 1.44$ | $-0.16 \pm 1.27$ | 0.873 | – |
| #6a | WRF/MYJ + ODIAC | 75 | 16 | 0.05 | $0.30 \pm 1.31$ | $-0.06 \pm 1.22$ | 0.989 | 0.830 |
| #6b | WRF/MYJ + ODIAC | 65 | 25 | 0.1 | $0.30 \pm 1.31$ | $-0.06 \pm 1.23$ | 0.959 | 0.826 |

*Data from Sodegaura on 23 March 2016 were excluded.



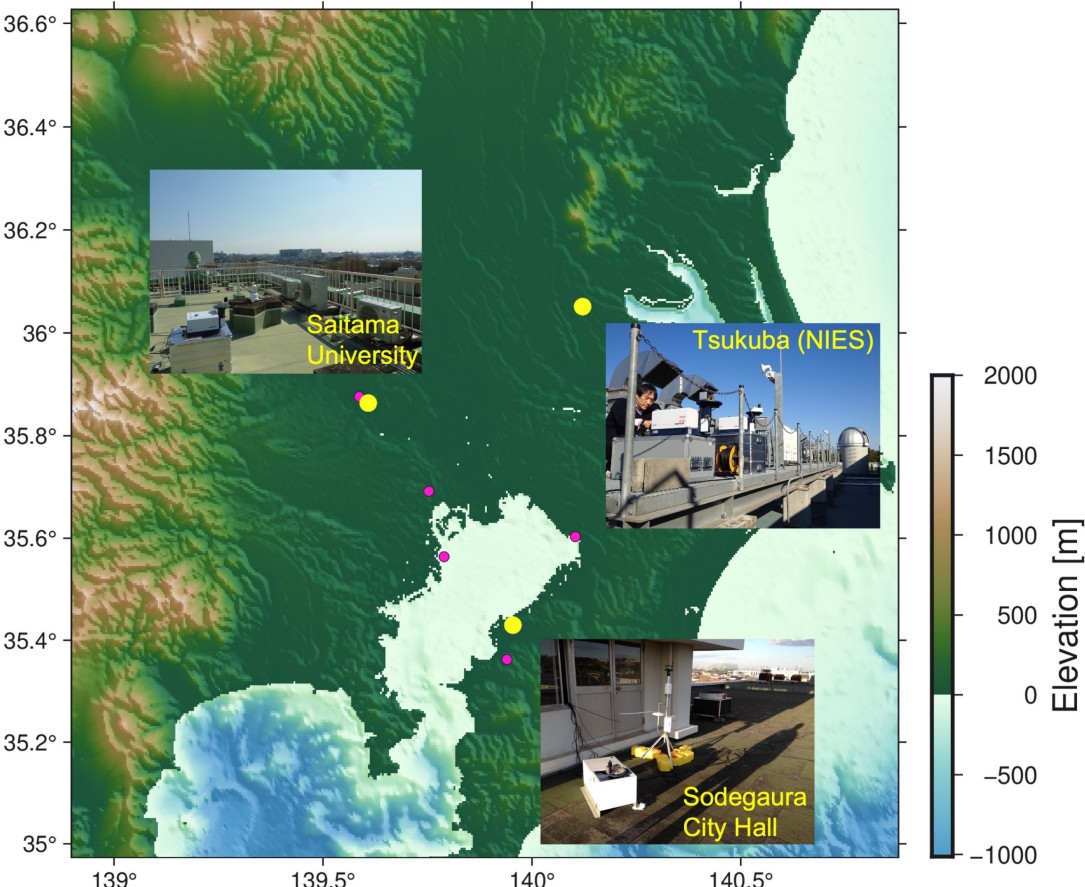

**Figure 1.** Locations of the two EM27/SUN observation sites (Saitama University and Sodegaura City Hall) and Tsukuba
TCCON site (yellow circles). Also shown are the AMeDAS stations (red circles) used for the comparison with wind data from
the WRF simulation and the ERA5 reanalysis data. The calculation of the footprint by WRF–STILT was performed for the
entire region displayed in this figure. The elevation data are from the Global Bathymetry and Topography at 15 arcsec
(SRTM15+ V2.1) (Tozer et al., 2019).




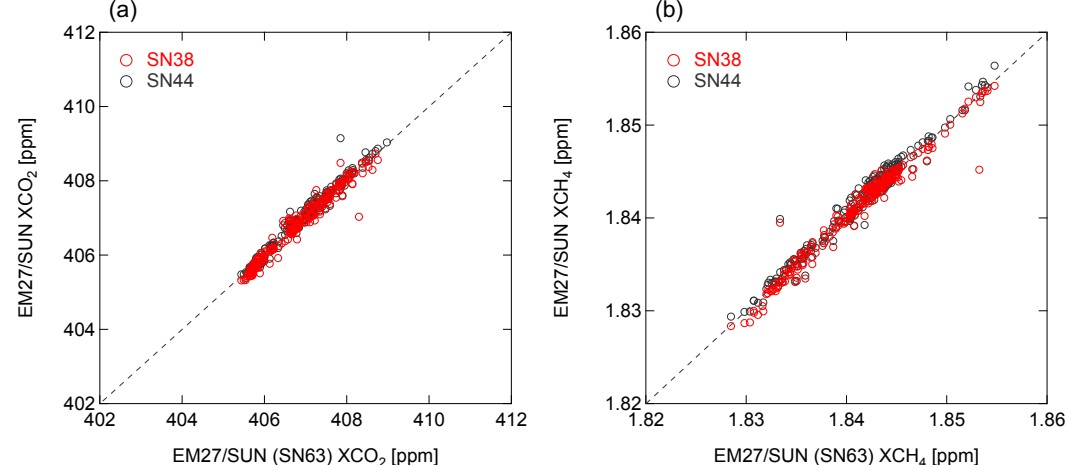

**Figure 2.** Scatter plots of bias-corrected SN38 and SN44 EM27/SUN data with respect to the SN63 EM27/SUN data for (a) $XCO_2$ and (b) $XCH_4$. Dashed lines denote the one-to-one line.



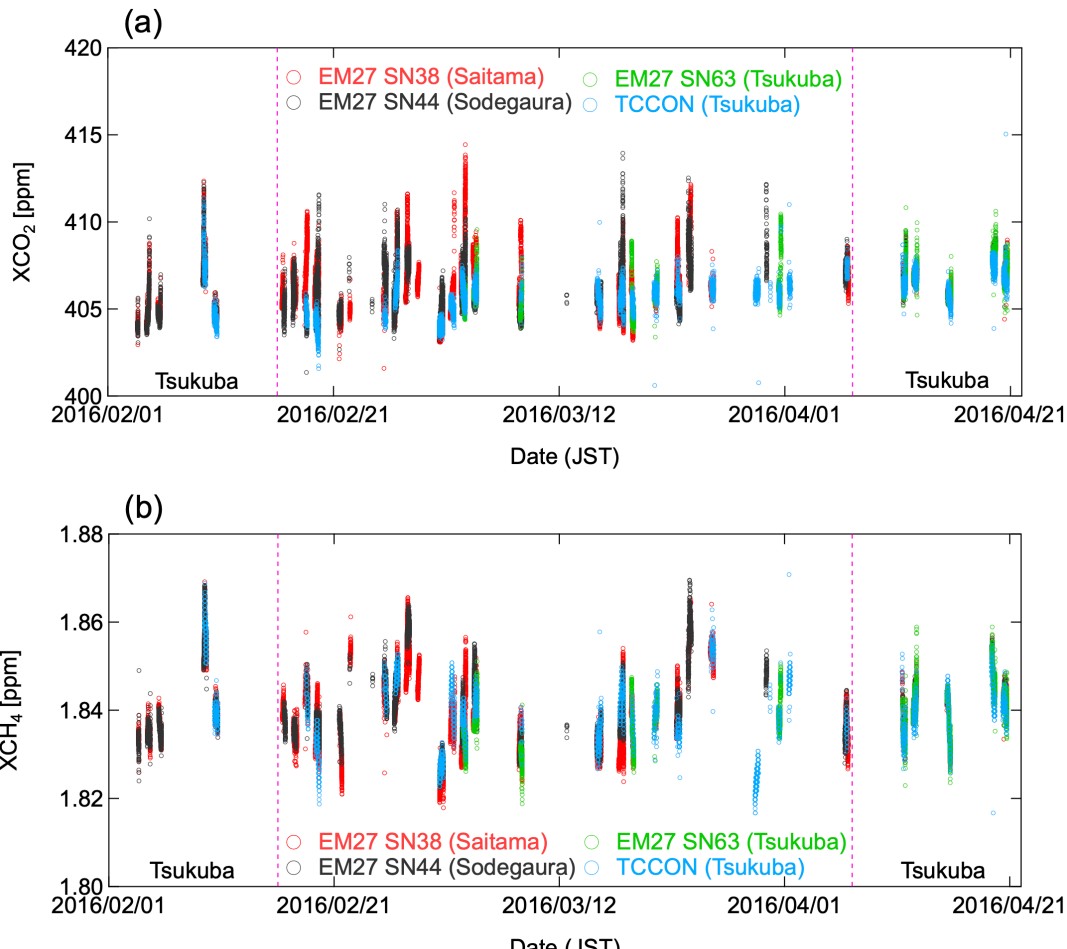

**Figure 3.** Time series of XCO₂ and XCH₄ during the observation campaign, including side-by-side measurements conducted at Tsukuba. The dashed vertical lines show the dates when the field observations began and ended at the three sites.



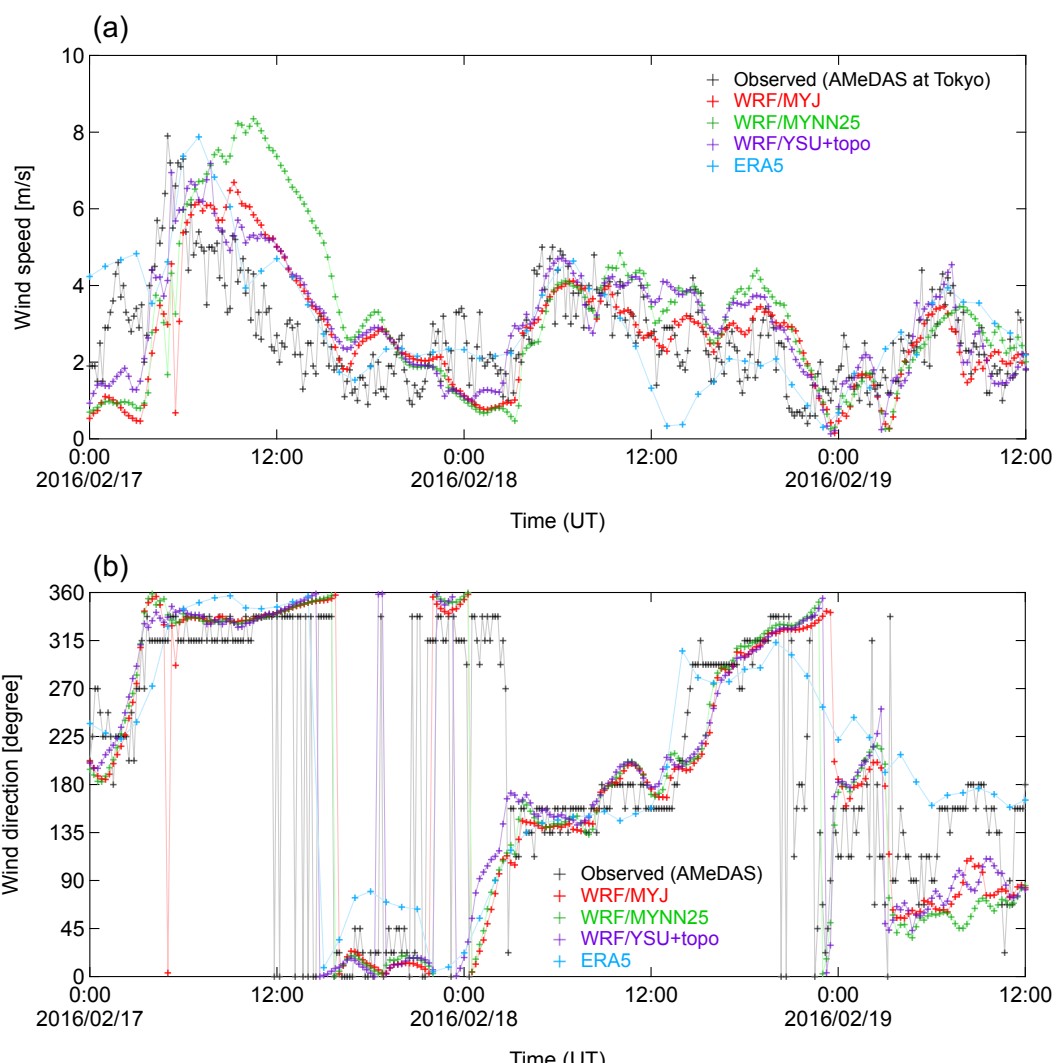

**Figure 4.** (a) Wind speed and (b) wind direction at the Tokyo AMeDAS station. The model data from the WRF simulations
using different PBL schemes and the ERA5 reanalysis are also shown.



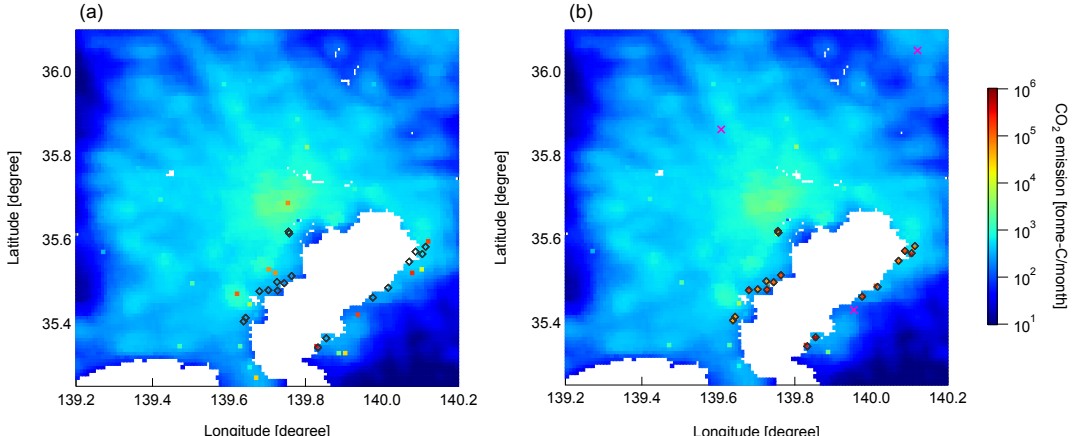

**Figure 5.** Anthropogenic $CO_2$ emissions from the TMA in March 2016 in (a) the original ODIAC 2020b data and (b) the same data except that the locations and emission magnitudes of large point sources, such as power plants and manufacturing plants, were corrected based on the national emission inventory. Open diamonds denote the locations of large point sources, and the crosses in (b) denote the observation sites.



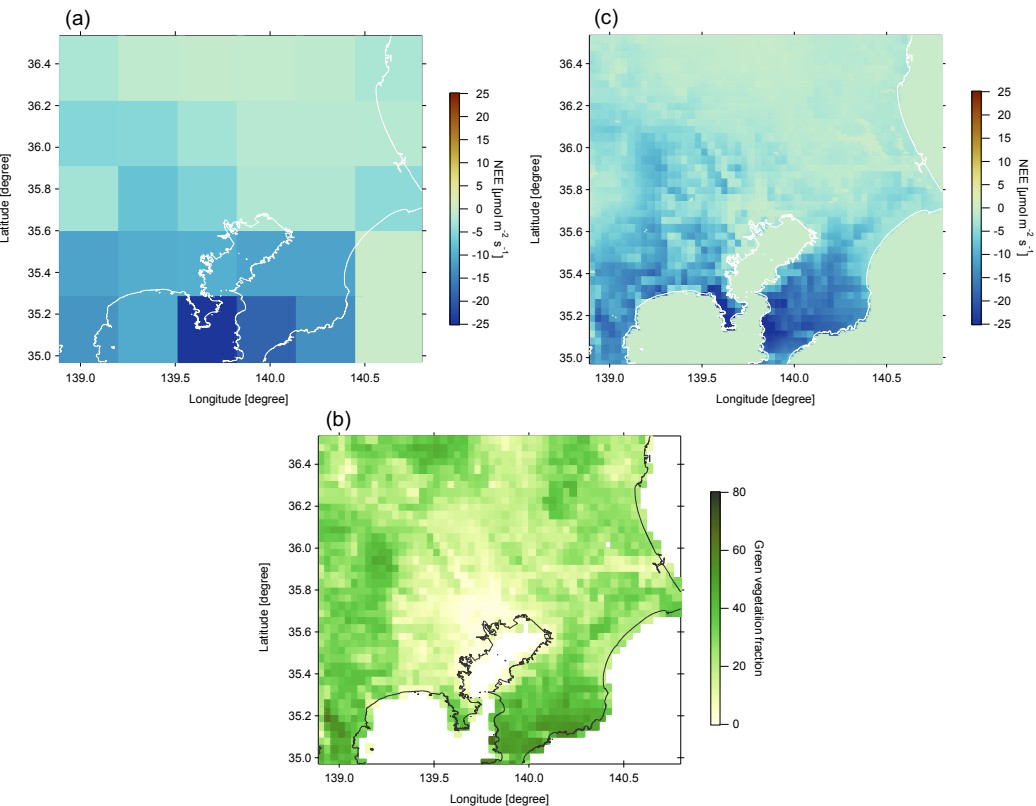

**Figure 6.** (a) NEE data at 03 UTC 23 March 2016 from the VISITc model and (b) GVF data during 20–26 March 2016. (c) NEE data downscaled using the GVF data.



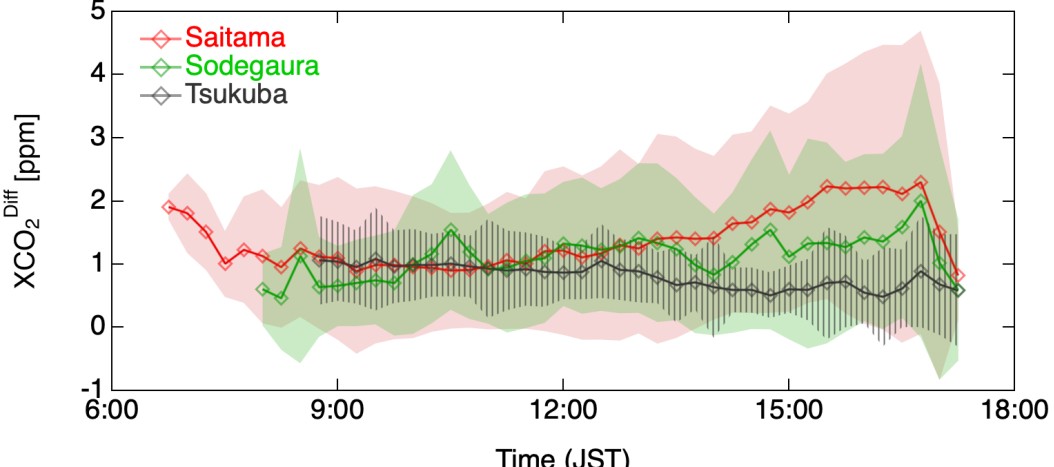

**Figure 7.** Average diurnal variations in $XCO_2$ differences ($XCO_2^{Diff}$) from daily background values. These background values
were assumed to be common to the three sites and be the 5th percentile value of the Tsukuba TCCON measurements throughout
each day. The average $XCO_2^{Diff}$ values (open diamonds) and their standard deviations (shading) were calculated for 15-min
bins using all data acquired during the campaign period.



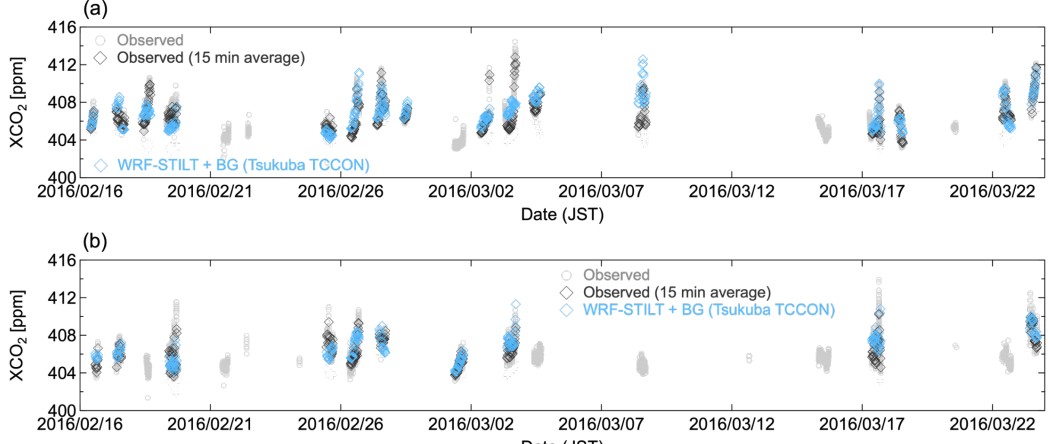

**Figure 8.** Comparison of the XCO₂ observations with the WRF–STILT simulation results (open blue diamonds) at (a) Saitama and (b) Sodegaura. The observations are presented as individual values (open gray circles) and as the 15-min averaged values used for the inversion (open black diamonds).





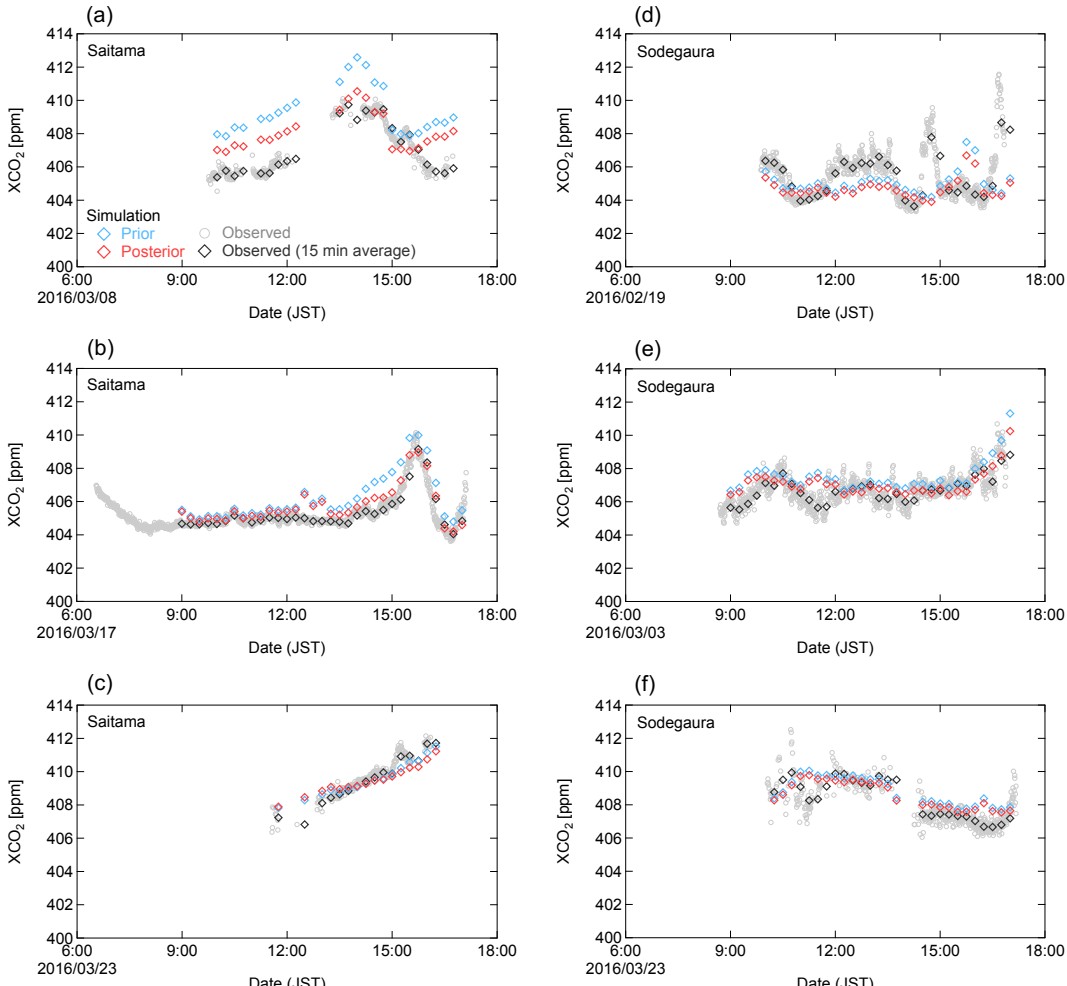

**Figure 9.** Comparison of the XCO₂ observations with the WRF–STILT prior (blue) and posterior (red) simulation results for three representative days at (a–c) Saitama and (d–f) Sodegaura. The observations are presented as individual values (open circles) and as the 15-min averaged values used for the inversion (open diamonds).



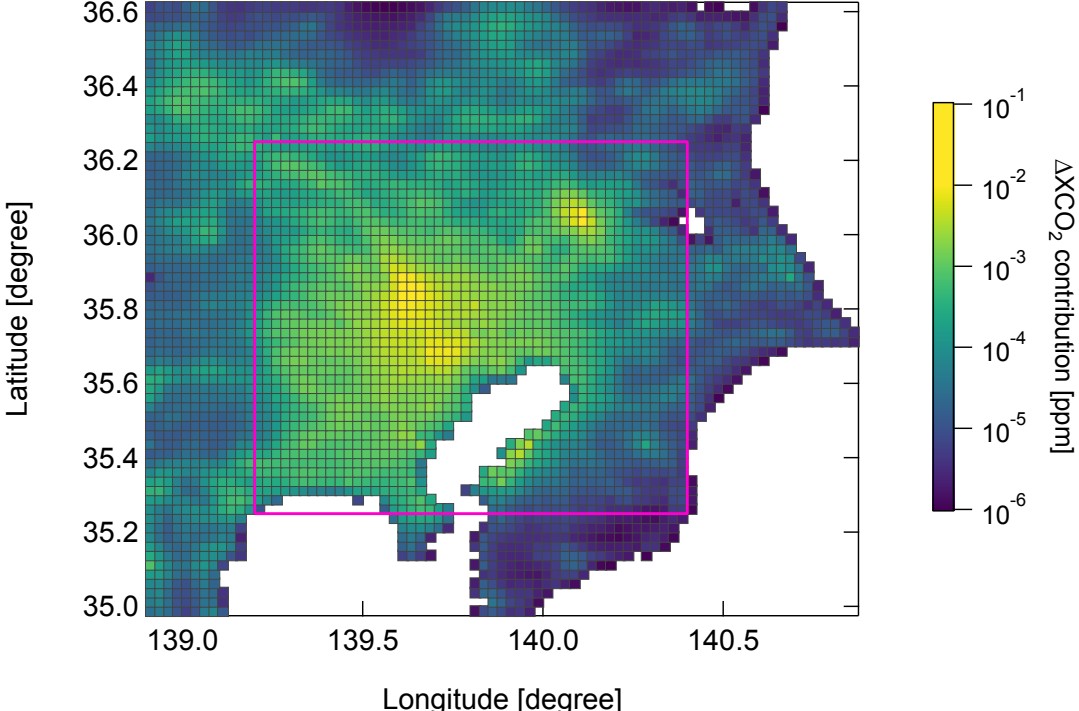

**Figure 10.** Mean contribution of each grid cell to the $\Delta XCO_2$ values simulated from prior nonpoint source emissions over the campaign period. The $CO_2$ emissions were optimized for the domain within the magenta rectangle.



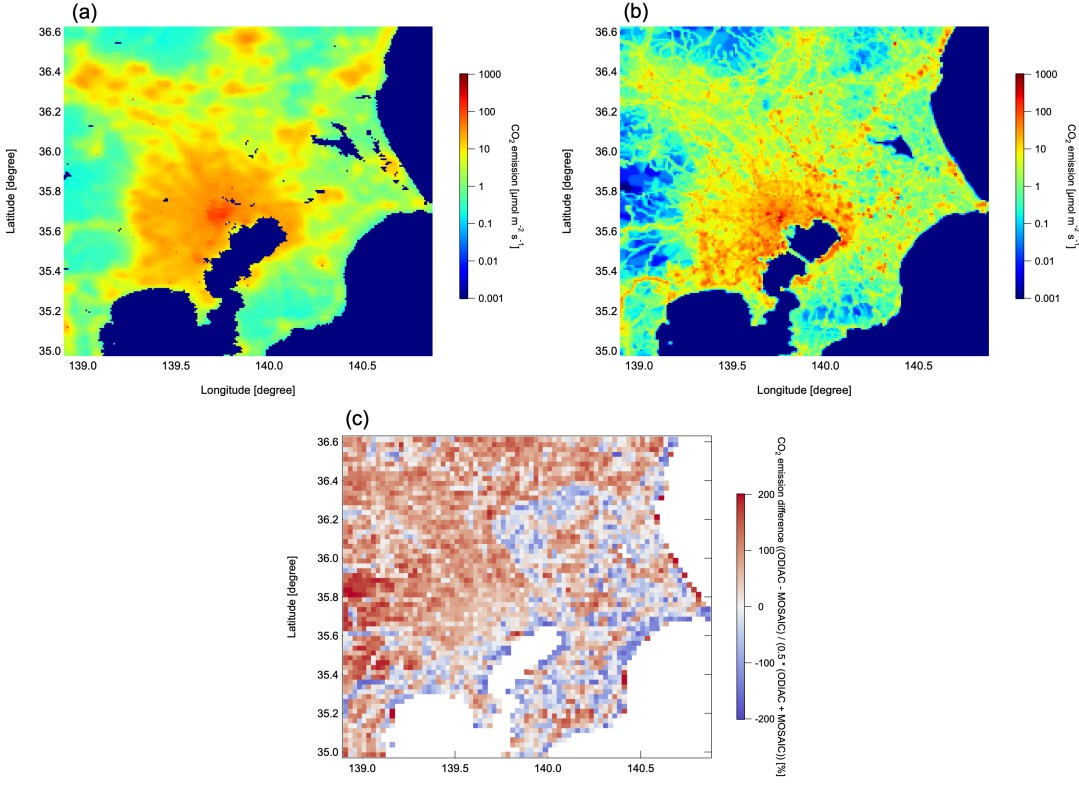

**Figure 11.** Average CO$_2$ emission fluxes from (a) ODIAC2020b data in February and March 2016 and (b) MOSAIC data in February and March 2015. (c) The difference between the two datasets aggregated to 0.025° × 0.025° spatial resolution, calculated as (ODIAC – MOSAIC) / (0.5 × (ODIAC + MOSAIC)) × 100. Note that large point sources have been excluded.



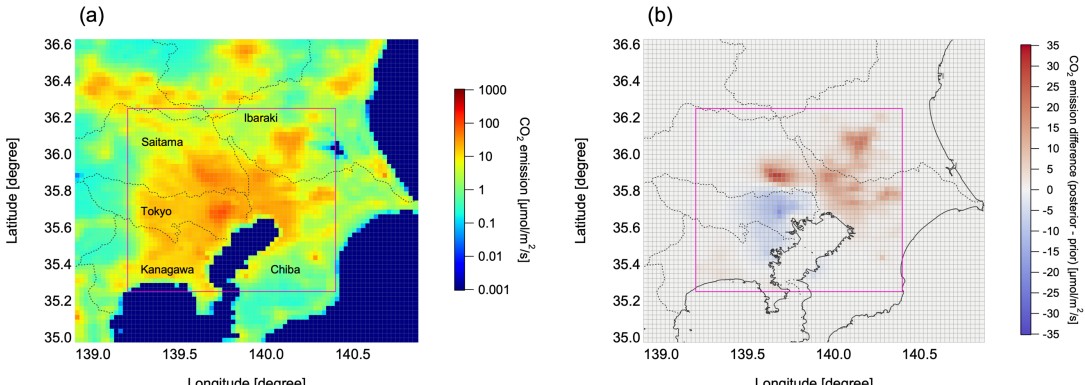

**Figure 12.** (a) Nonpoint source $CO_2$ emission fluxes in the TMA in the reference inversion (case #0 in Table 5) combining posterior (within the magenta rectangle) with prior (outside the rectangle) $CO_2$ emissions and (b) the difference between the posterior and prior emissions. The dotted lines show the administrative boundaries.





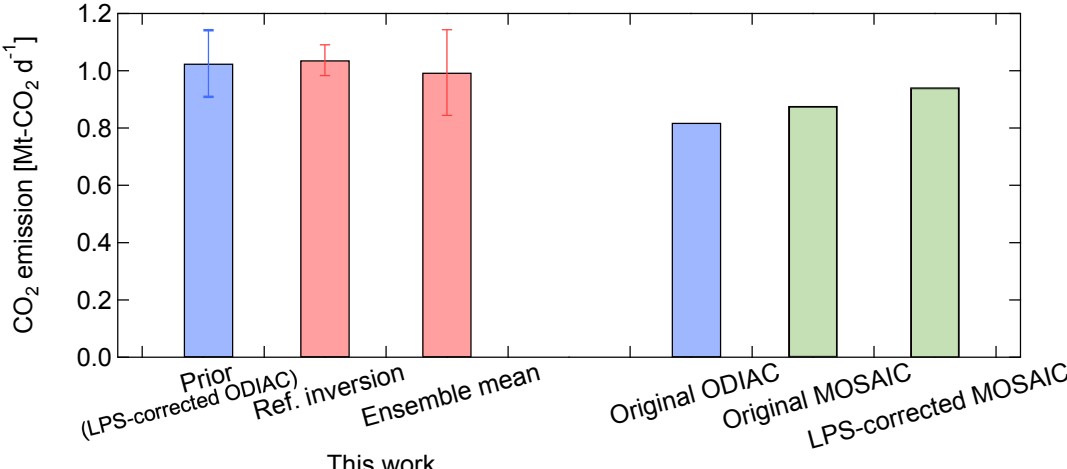

**Figure 13.** Total $CO_2$ emissions in the TMA calculated from the posterior emission fluxes (red), ODIAC data (blue), and MOSAIC data (green). The posterior emission fluxes are shown for the reference inversion (case #0 in Table 5) and the ensemble mean of all cases listed in Table 5. The error bars for the prior and posterior emission fluxes are the respective estimated uncertainties, whereas that for the ensemble mean is the standard deviation. For the ODIAC and MOSAIC data, both original and LPS-corrected total emissions are shown.