# Peer review of "Anthropogenic CO2 emission estimates in the Tokyo Metropolitan Area from ground-based CO2 column observations"

_EGUsphere, 2023_

## Referee Comment (RC2)

The authors developed an inversion scheme to infer the anthropogenic carbon dioxide emissions in the Tokyo Metropolitans Area from observations of three ground based remote sensing sites. One of which is a TCCON site.

The authors obtained the background by subtracting the simulated $CO_2$ enhancement (from the footprint and surface flux) from the observed $XCO_2$ values at the Tsukuba COCCON site for the forward modeling. To assess the biosphere, they spatially downscaled the terresstrial biospheric model VISITc to simulate the biogenic influence and found the influence to the enhancements to be small. The authors infer the meteorological surface interaction using WRF-STILT with a spatial resolution of 1km. The Bayesian inversion scheme inverts for spatially resolved emissions, separated into point and area-sources for the more than two months period with a total of approximately 6.5 degrees of freedom. The authors also compared 12 different model configurations.

The authors report total carbon dioxide emissions for the study area and compare it to several literature reports and find good agreements within the reported uncertainties. The scientific value is to be rated as high, since emission estimates from observations still remain a tough challenge and needed to confirm or refine reported emission inventories. The paper is written in a clear, structured style. However, some details need improvements.

Potential weaknesses are:

1. The authors conduct inversion in log-space, and therefore negative emissions are suppressed, which is not very realistic. The biogenic model needs to be perfect, so that we can be sure that there are no "negative emissions".
2. DOFS of 6.49 implicates that solution tend to stick to the a-priori, given that the dimension of the state vector is rather large (m = 1921 or 481 or 121).
3. The authors assumed that all the sites have the same background air. It is not always true, when considering the transport time that the air needed to travel from upwind to downwind especially when the distance between the sites are big (~ 60 km).
4. The definition of background is confusing. The authors have two definitions of background in the paper, i.e. 5 percentile value of TCCON station at Tsukuba and observed $XCO_2$ from Tsukuba COCCON site subtracted with simulated $CO_2$ enhancement.

I would appreciate if the authors could comment on their thoughts on the potential weaknesses and/or discuss it in the paper, before the acceptance.

Detailed comments:

L 15: Suggestion: "We conducted ..." --> "In order to infer a top down emission estimate, we conducted..."

L17: I thought that you deployed 3 EM27SUN spectrometers, please clarify.

L 22: "nonpoint source" --> I would suggest the term "area source" (29 occurrences)

L 26: "emission fluxes at > 3km" To my understanding, the WRF-STILT resolution is 1km, please clarify.

L 31: Please add your final emission number for the study area, or at least the scaling factor with the according uncertainty to the abstract and if feasible, compare it to the literature references.

L 87: Suggestion: "when the daily sunshine duration in this region is high" --> "during the high-insolation period" for clarity and specificity.

L 93: "city center" --> "city-center"

L101: "ASL" --> "a.s.l." (standard abbreviation, multiple occurrences)

L107: " and is now continuously operated " --> " and has since been continuously operated "

L116: "interval of approximately 1 min" --> "interval of about 1 minute"

L130: What is the integration time for determining sigma? If it is 1 min, it might be useful to also report the 15 min values as you did for comparing the observations with the forward simulations.

L132: you scale the TCCON to EM27, would it not make more sense to scale EM27 to TCCON, since TCCON is considered as standard.

L135: Maybe mention the altitudes of the stations somewhere in the text

L142: unit wrong → ppm/(mol/m²/s)

L145: Since you use the exact same altitudes as T.S.Jones et al.,2021 uses, you can add a citation here.

L154: "multiplied by anthropogenic and biogenic fluxes" --> "multiplied with spatially resolved emission inventories for anthropogenic and biogenic fluxes separately"

L155ff: "The change ... over all grid cells." --> "The change ... over all grid cells and serves for the forward modeling."

L150ff: "We then aggregated the footprints in each grid over the STILT run time." It is not clear what you mean by "aggregate". If it is meant as an introduction into the following sentence I would suggest to move the line break before this sentence.

L243: please break down this long sentence into at least two shorter ones

L248: "the original VISIT" --> "the initial VISIT"

L251: Gaussian T382 Grid --> please explain shortly, give reference or just state something like "operate on exactly the same grid" in order to make your point.

Section 3.3 in general: Multiple sentences are very long; consider breaking them into shorter pieces for a better understanding.

L257: How you can downscale VISITc product from 0.31 * 0.31 deg. to 1 km x 1 km using 4 km resolution GVF data? I am not sure whether you have the high-resolution information necessary to achieve this goal.

L267: "DXCO2 values measured" This statement is confusing, since DXCO2 values are derived from the forward model as described in the referenced section 3.1

L268: "H, representing atmospheric transport" --> To my understanding it is the forward model.

L273: Is it correct, that you have the logarithmic of a scaling factor (unitless) as well as an emission value (in mole/area/time) in the state vector x? Please clarify.

L279: Inverting in the log-space introduces a strong bias to positive emissions. Negative emissions are not necessarily non-physical, especially in case of CO2, because biospheric activity might be stronger than assumed. Did you try to invert in linear space? Negative emissions could serve as a sanity check here.

L331: "sources, large point sources" --> "sources, strong point sources" to separate from the spatial meaning of "large"

L311ff: It is not very clear in the text what XCO2^{Diff} means and how it separates from (DXCO2).

L315: It is a bit confusing here, because you define another background (5 percentile value of the Tsukuba TCCON site) than the one you use for the forward modeling. What XCO2^{Diff} is actually used for? Just to look into temporal fluctuation?

L312: Please explain the reasons to use Tsukuba as a backgound site.

L314ff and L355: How many days (or observations of the n=654 observations) were replaced by CarbonTracker?

L359: you averaged the data in 15 mins. Why is it optimal or in another word, why no drift of the sensor is integrated? You could refer to: https://acp.copernicus.org/articles/16/8479/2016/acp-16-8479-2016.pdf, section 3.1, where the optimal integration time is determined by using Allan analysis.

L370: You talked about the model-observation discrepancy, forward modeling vs. observation is mainly given by the errors in the WRF-STILT, what about the background error?

L445: you are looking into the model-observation mismatch for the inverse modeling framework. However, in your inversion you assume the same background for all sites. The background influence is canceled out in the forward model. Why you need to take the uncertainty of the background into account?

L455: It is not exactly clear what the authors mean with "upward" and "downward"

L470: With 6.5 degrees of freedom the model has not enough freedom to scale the sources individually. What happens if you provide an intentionally much uncertain a-priori (e.g. Factor 2 higher).

L475ff: Table 5: Please add the degrees of freedom and the Bayesian Information Criterion (BIC) to this list. The latter is a helpful number to tell which of the models could be a better choice.

L494ff: The statement appears reasonable. However, referenced Fig. S6 does not appear to have a connection to this statement.

L497ff: The sentence is very long. Please reformulate.

L526: I thought a third EM27/SUN is also deployed at Tsukuba site.

L553: Again here is 3km resolution mentioned. To my understanding it is 1km. If not correct please explain the reasons.

L915: "sigma_a" for prior uncertainty instead of "sigma_e".

General model description:

lack of overview and strict separation of description of the inversion methodology, model setup details and results

---

## Author Comment (AC1)

Referee #1

Overall comments:

The paper is packed with useful information, and it seems the authors have invested tremendous effort.
While some details are missing, making certain parts challenging to follow, the prior simulation itself
is commendable. I believe this paper is suitable for publication in EGUsphere once the authors address
the comments.

I would like to suggest that the authors dedicate some time to refining the sentence structures for a 10 smoother reading experience. Additionally, as mentioned below, I recommend relocating certain 11 paragraphs from the Results section to the Methodology section or the supplementary materials, as the 12 two sections appear to be mixed.

Furthermore, I have a specific request regarding Figure S5: It would be beneficial to include a scatter plot comparison that depicts "local" enhancements by subtracting the background. I am curious about how the background estimation was carried out and affects the scatter plot comparison. Additionally,

- 17 I am curious to know whether the inversion was performed after the background subtraction.
- 18

I hope that the authors will thoroughly address the detailed comments below.

We thank you for your careful reading of our paper and for providing your valuable comments. We have refined the sentence structures, clarified the treatment of background, and revised our manuscript
 according to your comments. Please see our specific responses below.

Regarding Figure S5 (Figure 9 in the revised manuscript), a scatter plot between the simulated XCO2

enhancements ( $\Delta XCO_2$ ) and the observed  $\Delta XCO_2$  after subtracting the background is shown in Figure

R1a, together with a scatter plot for XCO2 (Figure R1b, same as Figure 9 in the main text). The background is defined as the Tsukuba TCCON XCO2 data minus the simulated Tsukuba XCO2

enhancements.

In the initial manuscript, the differences in XCO2 measurements between the urban and Tsukuba sites were considered observational data and the corresponding XCO2 differences were simulated (as represented by Equation (7) in the initial manuscript). However, in Figures 8 and 9 (in the initial manuscript), the observed and simulated "XCO2" were shown by transforming the equation. This may have caused some confusion. In the revised manuscript, the XCO2 measurements at the urban sites have been considered observational data (as represented by Equation (2) in the revised manuscript) to

- 35 be consistent with what those figures show. We note that this change is mathematically identical in the
- 36 inverse analysis (just movement of a few terms in the equation) and does not affect the inversion

**37 results at all.**

Figure R1. (a) Scatter plot between the XCO2 enhancements ( $\Delta$ XCO2) simulated from the prior (black) and posterior (red) emission fluxes and the observed  $\Delta$ XCO2. (b) Scatter plot between the simulated and observed XCO2 values. The mean difference between the simulations and observations (simulation minus observation) with the standard deviation ( $\pm 1\sigma$ ) is denoted as  $\delta$ , and *r* is the correlation coefficient.

Detailed comments:

L28-29: The following statement is subjective because it depends on the a priori assumption. For 49 example, if the prior is assigned with large uncertainty, the percentage of uncertainty reduction in the 50 posterior will be larger, e.g., even larger than a factor of 3. So, the author needs to clarify this sentence: 51 "In addition, the inverse analysis reduced the uncertainty in total CO2 emissions in the TMA by a 52 factor of  $\sim$ 2."

- 53 We have revised the last two sentences of the abstract as follows: "The prior and posterior total CO2 54 emissions in the TMA are  $1.026 \pm 0.116$  and  $1.037 \pm 0.054$  Mt-CO2 d-1 at the 95% confidence level, 55 respectively. The posterior total CO2 emissions agreed with emission inventories within the posterior
- 56 uncertainty, demonstrating that the EM27/SUN spectrometer data can constrain urban-scale monthly
- 57 CO2 emissions."
- 58

L30-31: Instead of the current conclusion, I recommend the authors use a statement, e.g., the posterior

- 60 emissions are X+/-Y times the prior emissions (at the 95% CI). This way, the readers get more
- 61 information, e.g., how tightly the measurements constrain the emissions.
- 62 We have revised the sentence as mentioned above.

- 63
- 64 L76 78: I strongly recommend that the authors add a couple of sentences describing this work's
unique contribution in addition to the previous work for the TMA.
- 66 We have added the following sentences: "We constructed CO2 emission inventories with more accurate
- 67 information on both the locations and emissions of large point sources. Anthropogenic CO2 emissions
- 68 from area sources and large point sources were estimated separately using this inventory as the prior.
- 69 In addition, the area source emission estimates with higher spatial resolution allow verification of the
- 70 emissions reported by each administrative division."
- 71
- 72 L90: I would recommend that the authors add a map of Japan as an inset to show the relative location
- of the study area. The elevation map is good, but it is hard for those unfamiliar with the area to make
- sense of the study area relative to the entire country.
- 75 We have added a map of Japan to Figure 1. In addition, we have added the following sentence to the
- caption of Figure 1: "The upper right figure shows the location of the study area relative to Japan as awhole."
- 78
- L142: Was the footprint normalized? The unit for footprint should be "ppm/flux" or, specifically, "ppm
   /(umol/m2/s)"? It seems that clarification is needed.
- 81 We have corrected the unit for the footprint to " $ppm/(\mu mol/m^2/s)$ ".
- 82
- L258: Are the authors referring to the GVF data from VIIRS? It would be useful to add the exactVIIRS product name.
- 85 We have revised the sentence as follows: "we spatially downscaled the hourly VISITc NEE data using
- 86 GVF data from the Visible Infrared Imaging Radiometer Suite (VIIRS) sensor onboard the Suomi
- 87 National Polar-orbiting Partnership satellite (VIIRS Global Green Vegetation Fraction). The GVF data
- 88 are produced with an approximately 4-km spatial resolution on a daily basis from the past 7 days of
- 89 VIIRS observations (Ding and Zhu, 2018)."
- 90
- 91 L263: As written, it is not clear. Was the ratio of the interpolated GVF versus the original GVF applied
- 92 to the NEE data at 1 km? Or something else?
- 93 We have revised the sentence as follows: "The ratio of the original GVF to the interpolated GVF was
- 94 multiplied by the interpolated NEE data to produce the downscaled NEE data (Fig. 4c)."

- 96 Section 3.3: Overall, I think the authors did a good job of making the prior fluxes more accurate!
- 97 Thank you for your positive feedback.
- 98

L264 – 265: Suggestion for rewriting to improve clarity: "The downscaling process was conducted in
a manner that ensured all original sums of the NEE data from the TMA were preserved following the
downscaling."

- 102 We have made this revision.
- 103

L268: "forward" seems wrong. First, WRF-STILT is not a physical "forward" model in this setting,
although it can be used for forward simulation. Second, this is a linear or nonlinear model, statistically
speaking.

As you pointed out, the WRF-STILT simulations were performed in "backward" mode to trace back 108 the origin of the observed airmasses. However, in the present study, the terms "forward simulation" 109 and "forward model" mean the process for calculating XCO2 values from surface fluxes via 110 atmospheric transport as opposed to an "inverse analysis" or "inverse model" that infers surface fluxes 111 from XCO2 values. In the revised manuscript, we have explicitly written the forward model (Equations 112 (2) to (4)). In addition, we note that the term "forward simulation" is also used in other similar studies 113 on top-down emission estimates (e.g., Cusworth et al., 2020; Huang et al., 2019; Maksyutov et al., 114 2021; Pisso et al., 2019).

Cusworth, D. H. et al.: Synthesis of methane observations across scales: Strategies for deploying a
multitiered observing network, Geophys. Res. Lett., 47, e2020GL087869,
https://doi.org/10.1029/2020GL087869, 2020.

Maksyutov, S. et al.: Technical note: A high-resolution inverse modelling technique for estimating
surface CO2 fluxes based on the NIES-TM–FLEXPART coupled transport model and its adjoint,
Atmos. Chem. Phys., 21, 1245–1266, https://doi.org/10.5194/acp-21-1245-2021, 2021.

Huang, Y. et al.: Seasonally resolved excess urban methane emissions from the Baltimore/Washington,

- 123
   DC
   Metropolitan
   region,
   Environ.
   Sci.
   Technol.,
   53,
   11285–11293,

   https://doi.org/10.1021/acs.est.9b02782, 2019.
- 125 Pisso, I. et al.: Assessing Lagrangian inverse modelling of urban anthropogenic CO2 fluxes using in
- 126 situ aircraft and ground-based measurements in the Tokyo area, Carbon Balance Manage., 14, 6,
- 127 https://doi.org/10.1186/s13021-019-0118-8, 2019.
- 128
- 129 L269: I suggest that the authors present H(x, b) more explicitly, e.g., by writing out the Jacobian matrix
- 130 and x together. That way, the reader can understand the nonpoint and point source inversion more
- 131 easily. This is related to Eq. (3), where "K" is introduced. Showing how "K" is associated with "b"
- 132 should be useful (unless it is presented in the supplemental; I don't see it).
- 133 We have added the following sentences: "The forward model simulates XCO2 values at the urban sites
- 134 (Saitama or Sodegaura) as follows:

$$H(\mathbf{x}, \mathbf{b}) = \Delta XCO_2 \operatorname{strikt}_{2}(\mathbf{x}, \mathbf{b}) + XCO_2 \operatorname{BG}(\mathbf{x}, \mathbf{b}), \qquad (2)$$

where  $\Delta XCO_2 \frac{\text{urban}}{\text{STILT}}$  is the XCO2 enhancement at the urban sites simulated by the pressure-weighted 137 footprint and the surface fluxes, and  $XCO_2^{BG}$  is the background value. We calculated the  $\Delta XCO_2$ 138 values as follows:

$$\Delta XCO_{2 \text{ STILT}}^{\text{urban}}(\boldsymbol{x}, \boldsymbol{b}) = \boldsymbol{F}_{\text{aggr}}^{\text{urban}} \boldsymbol{x}_{\text{area}} + \boldsymbol{F}_{\text{fine}}^{\text{urban}} \boldsymbol{b}_{\text{point}} \boldsymbol{x}_{\text{point}} + \boldsymbol{F}_{\text{fine}}^{\text{urban}} \boldsymbol{b}_{\text{bio}}, \qquad (3)$$

where  $F_{\text{fine}}$  and  $F_{\text{aggr}}$  are the original and the spatially aggregated footprints, respectively.  $x_{\text{area}}$ 141 and  $x_{\text{point}}$  are the emission flux vector for area sources and the (scalar) scaling factor for large point 142 sources, respectively.  $b_{\text{point}}$  and  $b_{\text{bio}}$  are the emission flux vectors for large point sources and 143 biogenic sources, respectively."

"We therefore obtained the background  $XCO_2$  values by subtracting the simulated  $\Delta XCO_2$  values at (4)"

the Tsukuba site ( $\Delta XCO_2 \frac{Tsukuba}{STILT}$ ) from the Tsukuba TCCON XCO2 values (XCO2  $\frac{Tsukuba}{TCCON}$ ):

$$XCO_2^{BG}(\boldsymbol{x}, \boldsymbol{b}) = XCO_2^{Tsukuba} - \Delta XCO_2^{Tsukuba}(\boldsymbol{x}, \boldsymbol{b})$$

$$= XCO_2^{Tsukuba} - (\boldsymbol{F}_{aggr}^{Tsukuba} \boldsymbol{x}_{area} + \boldsymbol{F}_{fine}^{Tsukuba} \boldsymbol{b}_{point} \boldsymbol{x}_{point} + \boldsymbol{F}_{fine}^{Tsukuba} \boldsymbol{b}_{bio}),$$

Also, it is not clear at which temporal resolution the authors solve for "x." Are you solving for sub-150 daily emissions for each pixel? Yes, is it also solved for each pixel as well? If so, how the "b" matrix 151 is constructed? I am asking this question because the authors use hourly emissions, at least for NEE 152 and anthropogenic. Then the "b" matrix should be extensive. As it is written, many things are not clear. 153 The state vector x was optimized as a single average during the entire campaign period. The temporal 154 variation of anthropogenic emissions (weekly and diurnal correction factors from the TIMES model) 155 was taken into account in summing the hourly footprints over the STILT run time. On the other hand, 156 x consists of only "average" area source emission fluxes from ODIAC for each pixel, and such a single 157 set of fluxes were optimized. The hourly biogenic fluxes are all included in *b*.

We have revised the two descriptions on the application of the TIMES model as follows:

(1) "The hourly footprints calculated over the STILT run time (24 h) at a given time were weighted by temporal correction factors of CO2 emissions (described in Sect. 3.3) and aggregated in each grid cell."
(Section 3.1)

(2) "Because we applied weekly and diurnal correction factors from the TIMES model to the hourly
footprints in summing them over the STILT run time, we optimized one static emission distribution
during the campaign period, assuming that the temporal variation of the emissions followed the
TIMES model." (Section 3.5)

In addition, we have added the following sentence in Section 3.5: "Similarly, a single average scaling

- 167 factor for the large point sources was optimized from the data over the entire campaign period."
- 168
- 169 L270: I would not recommend using "state" in the fixed quantity as in the sentence "b is the fixed state
- 170 vector"; "State" is typically suitable for parameters (please change accordingly if "state" was used for

"b." in other places)

| 172 | We have revised the sentence as follows: " $\boldsymbol{b}$ is the vector consisting of fixed physical quantities." |
|-----|---------------------------------------------------------------------------------------------------------------------|
|     |                                                                                                                     |

L272-273: Based on "the state vector x includes spatially resolved nonpoint source emissions and a 175 scaling factor of the large point source emissions," the reader may be confused about how the inversion 176 was done. Are you solving for the "flux" directly for the nonpoint source but the "scaling factor" for 177 the point source? If it is the case, it is ok. But it needs clarification. Maybe, the authors did this way, 178 but it is not clear from the writing.

- We did it the way you suggest. To clarify, we have revised the sentence as follows: "the state vector x
  includes spatially resolved fluxes for the area source emissions and a scaling factor for the large point
  source emissions."
- 182

L288: How is "the Levenberg–Marquart parameter" estimated? Or prescribed?

We have revised the sentence as follows: " $\gamma$  is the Levenberg–Marquart parameter fixed at 10 (Chen 185 et al., 2022)."

Chen, Z., Jacob, D. J., Nesser, H., Sulprizio, M. P., Lorente, A., Varon, D. J., Lu, X., Shen, L., Qu, Z.,
Penn, E., and Yu, X.: Methane emissions from China: a high-resolution inversion of TROPOMI
satellite observations, Atmos. Chem. Phys., 22, 10809–10826, https://doi.org/10.5194/acp-22-10809190 2022, 2022.

L314-315: I am curious how the authors matched the vertical profiles between CarbonTracker (CT) 193 and EM27 to get the background for EM27. A weighting scheme was used? Ideally, the particle 194 trajectory for each receptor (at different locations and vertical levels) of EM27 should be computed 195 and then averaged using a kernel (or a set of weights, likely based on pressure distributions) compatible 196 with EM27. To sample values from CT (using particle trajectories), the same method should be used 197 to match the vertical profile between the two. I wonder if the authors did that or something else.

We did not use the CO2 profile product of CarbonTracker, but rather the XCO2 product (CT2019B.xCO2), so we did not perform any weighing by the column averaging kernel. We have added the product name (CT2019B.xCO2), and these sentences have been moved to Section 3.4.

L310: By "XCO2 differences", do the authors mean "enhancement" above the background? The phrase "XCO2 differences (XCO2Diff) from daily background values" needs to be revised for clarification.

We have revised the sentence as follows: "To characterize the diurnal variation in  $XCO_2$  at each observation site, we examined the diurnal variation in  $XCO_2$  enhancements ( $XCO_2^{Enh}$ ) above the daily XCO2 baseline." We note that the "5 percentile value of the Tsukuba TCCON measurements" has been
referred to as the "baseline" in the revised manuscript, not to be confused with the "background"
defined and used in the simulations and inverse analyses.

L313: How did the author account for the background uncertainty based on this "5 percentile"assumption?

We have added the following sentences: "When the 2 (10) percentile values of the Tsukuba TCCON measurements were used as the daily XCO2 baseline, the maximum XCO2Enh values were 9.6 (9.4) ppm at Saitama and 9.5 (8.9) ppm at Sodegaura. These changes had little effect on the standard deviations of the mean XCO2Enh values and the pattern of the diurnal variation."

L318: Please add "diurnal" so that it reads "The average diurnal XCO2Diff." By the way, I think 219 " $\Delta$ XCO2" is more informative to represent the local signal (I find both are used). Some people use 220 "XCO2" to describe the local mixing ratio (after subtracting background). I suggest the authors review 221 the notation a bit more to avoid confusion. In fact, what is the difference between "XCO2Diff" and 222 " $\Delta$ XCO2" in Line 277? I may have misunderstood, but further clarification would help. Thank you.

We have revised the sentence as follows: "The average diurnal XCO2Enh value per 15-min bin was calculated for each site using the entire field campaign dataset (Fig. 6)."

" $\Delta XCO_2$ " is used only to represent the simulated local enhancements. " $XCO_2^{Enh}$  ( $XCO_2^{diff}$  in the initial 226 manuscript)" represents the observed XCO2 enhancements above the daily 5-percentile value of the 227 Tsukuba TCCON measurement. We have added the following sentence: "We note that the  $XCO_2^{Enh}$ 228 values were calculated using only the observed  $XCO_2$  values, whereas the  $\Delta XCO_2$  values represent 229 the simulations of local  $XCO_2$  enhancement."

L321: I am a bit confused to see that there is a moderate-level effect of biogenic fluxes while the
authors said, "the biogenic flux was allocated to the state vector b" in Line 276; it was assumed
negligible there. Any clarification?

The biogenic effect due to photosynthesis is not so large and the value is expected to be (relatively)

similar at the different sites. This is seen both for the observations (Figure 6) and simulations (Figure

S5). However, the biogenic fluxes are not small enough to be negligible, so they are included in the forward calculation of  $\Delta XCO_2$  (as the vector **b**). We have revised the sentences in Line 276 as follows:

"the biogenic flux was allocated to the fixed vector **b**. Note that the contribution of biogenic flux to the simulated  $\Delta XCO_2$  was small compared to that of anthropogenic flux and the differences among

ΔXCO2 calculated from four different biogenic flux products are also small (Sect. 4.2)."

L323: It is unclear what the authors mean by "the high early morning values at Saitama may reflect an airmass-dependent bias."

We have added the following sentences: "The airmass-dependent variation in  $XCO_2$  is caused by the effects of inaccurate spectroscopic parameters on the retrievals, which vary with the depth of the

- absorption lines (i.e., airmass) (Wunch et al., 2015). Although this effect is corrected in the GGG2014
- 247 software, the error may remain for a large airmass."
- 248

L330-334: The sentence sound awkward. Please revise.

We have revised the sentences as follows: "the XCO2 enhancement ( $\Delta$ XCO2) was calculated from the 251 column-averaged footprint and the surface fluxes from area sources, large point sources, and biological 252 activity. The  $\Delta XCO_2$  values resulting from the large point source emissions and biogenic fluxes were 253 calculated from the original footprints with a spatial resolution of approximately  $1 \text{ km} \times 1 \text{ km} (0.0083^{\circ})$ 254  $\times$  0.0083°). For area source emissions, however, we re-gridded the original footprints to a spatial 255 resolution of  $0.025^{\circ} \times 0.025^{\circ}$  to degrade the spatial resolution for the inverse analysis. First, the area 256 source emissions were summed for each  $0.025^{\circ} \times 0.025^{\circ}$  grid cell. Then, individual footprints for the 257  $0.025^{\circ} \times 0.025^{\circ}$  grid were derived by dividing the sum of the nine XCO2 contributions for the 0.0083° 258  $\times$  0.0083° grid by the emissions for the 0.025°  $\times$  0.025° grid."

L330-345: I would recommend that the authors move this particular paragraph to the Methodology
section or possibly to the supplementary materials. As it stands, the Results section seems a bit
extensive, and this adjustment could help with maintaining focus and flow.

We have moved L331-342 to the Methodology section.

L350-355: Here, the authors describe the background again, which I thought was done in Section 4.1. 266 Given that both mention "Tsukuba," I understood that site measurements were used as the background 267 common to the other sites. What's surprising to me is that the authors subtract the simulations at 268 "Tsukuba" from the "Tsukuba" measurements to remove the local enhancements for the background 269 site. It is possible, but it adds more uncertainty to the background because the simulated quantity itself 270 is uncertain. Typically, using the particle trajectories from the STILT model, we would sample 4-D 271 background data (over the ocean) simulated from a global model. The method used here is somewhat 272 convenient but adds uncertainty.

We believe that a method that takes the background from measurements away from the emissions is as typical as the method that combines the trajectory with the global model. In the present study, the

Tsukuba measurements were considered background due to their distance from the main emission

- sources. However, as demonstrated by Babenhauserheide et al. (2020), the Tsukuba measurements can
- 277 sometimes be impacted by emissions in the central area of the TMA. Therefore, the simulated
- 278 enhancements (from anthropogenic and biogenic emissions) at Tsukuba were subtracted from the

- Tsukuba measurements. As you pointed out, the simulated enhancements at Tsukuba added uncertainty
  to the background. However, optimizing the anthropogenic emission fluxes in the inversion analyses
  would reduce the uncertainties.
- 282
- Also, I suggest this paragraph be merged this the relevant paragraph in Section 4.1. Otherwise, the manuscript gets longer, and the reader is distracted/confused.
- We have refined the structure; this paragraph and the first paragraph of Section 4.3 have beencombined and moved to Section 3.4.
- 287
- L358-359: I suggest the authors add a scatter plot for predicted versus measured, corresponding to
  Figure 8, only for the 15-min average. I think the figures are already many, but Figure 2 and Figure 4
  (maybe more) can be moved to the supplemental.
- Such a scatter plot was shown as Figure S5 in the supplemental material. In the revised manuscript,
  we have moved Figure S5 to the main text (Figure 8). In addition, Figures 2 and 4 in the initial
- 293 manuscript have been moved to the supplemental section.
- 294
- 295 L360: By "the sum of the WRF–STILT  $\Delta$ XCO2 value every 15 min at each site and the background 296 XCO2 value", I assume " $\Delta$ XCO2" is the local enhancement. It needs to clarify between " $\Delta$ XCO2" 297 and "XCO2diff."
- 298 " $\Delta$ XCO2" is used only to represent the simulated local enhancements. "XCO2Enh (XCO2diff in the initial 299 manuscript)" represents the observed XCO2 enhancements above the daily 5-percentile value of the 300 Tsukuba TCCON site. In the revised manuscript, these have been clarified. We note, however, that this
- 301 sentence itself has been removed in the refinement of the sentence structure.
- 302
- 303 L361: "forward"? STILT back trajectories were used.
- 304 Although the STILT model was used in "backward" mode to calculate footprints, "forward" simulation
- 305 means the process to calculate XCO2 values from the footprints and the surface CO2 fluxes.
- 306 We have added a description of forward simulation in Section 4.2: "We compared the XCO2 data for
- 307 the forward simulations, which correspond to the XCO2 simulations from the footprints and the surface
- 308 CO2 fluxes based on Eqs. (2) to (4), with the EM27/SUN observations at Saitama and Sodegaura (Figs.
- 309 7 and 8)."
- 310
- 311 L363: What kind of point source? Is it identifiable, e.g., a power plant?
- 312 As shown in Figure 3 in the revised manuscript, there are several point sources near the Sodegaura
- 313 site, including steel plants as well as power plants, so it would not be possible to identify the source.
- 314 We have revised the sentence as follows: "which were likely caused by the plume from large point sources such as the power plants and steel plants located near the Sodegaura site."

L369: I don't necessarily agree with the statement: "Therefore, we attribute this large model-318 observation discrepancy to errors in the WRF-STILT model rather than to the emission data." First, I 319 don't expect ERA5 to perform better than WRF because it is a much coarse resolution model product 320 (I also see that in this work's Figure S3). From my experience, it can be much worse than WRF, 321 depending on the region. I would say that the authors only considered a limited set of meteorology, 322 not exploring a broader set of meteorological data. So, it is possible that the limited meteorology didn't 323 capture the temporal variation. However, as the author said, it is still possible that the short-term local 324 source not included in the prior fluxes is associated with this discrepancy between measurements and 325 predictions. To summarize, although it is likely that the transport source is the primary source of the 326 discrepancy, I don't see evidence for the strong statement above.

Since the simulations using the prior emission fluxes were able to reproduce the diurnal variation well, except for 3 March 2016, we thought that the modeling error on specific meteorological conditions might be the dominant cause of the discrepancy on that day. However, as you pointed out, a short-termlocal source not included in the prior fluxes could be the cause of the mismatch between the prior simulations and the observations on 3 March 2016.

We have revised the sentences as follows: "However, we cannot rule out the possibility that short-term local sources not included in the prior fluxes may cause the discrepancy between the prior simulations and the observations. Therefore, we attribute this large model–observation discrepancy to errors in the WRF-STILT model, or to the short-term local sources not included in the prior fluxes, or both."

L380: Equation 7 is confusing. What is the purpose of this equation? If this should be included, it should be presented in the section (e.g., 4.1) where the background is described. Based on the earlier description, wasn't *"XCOBG"* derived from *XCO*Tsukuba\_TCCON? As pointed out, this whole paragraph should be in the Method section, not the Result section.

In the revised manuscript, Equation (7) has been removed, and new equations that provide a detailed description of the forward model (Equations (2) to (4)) have been added. These equations make it clear that background is defined as the difference between the Tsukuba TCCON measurements and the

- 344 Tsukuba STILT simulations (i.e., Tsukuba TCCON minus Tsukuba STILT). In addition, this paragraph
- has been moved to Section 3.4.
- 346

L387-445: This should be included in the Method section for the abovementioned reason. There is no
 meaningful result described or discussed. They would agree with me if the authors read similar inverse
 modeling papers.

We have moved the description on the construction of the prior error covariance matrix and measurement error covariance matrix to the Methodology section (i.e., L387-412 in the initial manuscript). Because the remaining part (L413-446 in the initial manuscript) discusses the 352 353 uncertainties in our model-observation system based on the simulation results, it has been moved to 354 Section 4.2.

L417-418: This work differs from the system in Turner et al., where they have a dense measurement 357 network. I cannot offer any temporal correlation length scale for this work, but I am not quite sure 358 about adopting the 1-hr length scale.

Since Turner et al. (2020) have dense measurement data, a spatial correlation length scale and a temporal correlation length scale are imposed on the off-diagonal components of the measurement 361 error covariance matrix. The effect of the dense measurement data is taken into account by including 362 the spatial correlation length.

Meanwhile, as you note, the temporal correlation length is uncertain. We have added inversion 364 analyses using different temporal correlation lengths to the sensitivity analysis (in Section 4.3).

L448: Which period does Figure 12a represent? Is it the average of the hourly posterior fluxes during 367 the study period?

Figure 12a (Figure 11a in the revised manuscript) represents the single average emission fluxes 369 optimized using all data during the campaign period. As described above, this has been clarified in 370 Section 3.5 in the revised manuscript.

L470-471: Related to Equation 1, how many scaling factors were used/solved? Is this value of "0.856" 373 just the average of many scaling factors? A simple average of many scaling factors would not work, 374 though.

The scaling factor and the spatially resolved anthropogenic emission fluxes were each solved as single averages during the campaign period. We have added the following sentence to Section 3.5: "Similarly, a single average scaling factor for the large point sources was optimized from the data over the entire campaign period."

- 378
- 379

L512-513: Can the author offer further discussion on the difference between this study and Pisso et 381 al.?

We have added the following sentences: "Pisso et al. (2019) and this study use comparable Lagrangian transport models to calculate atmospheric transport; however, there are several differences, including

- 384 the type of observational data (in-situ vs. column), the prior emission fluxes (EDGAR vs. ODIAC),
- 385 the meteorological fields for driving the transport model (ERA-Interim vs. WRF based on GPV-MSM),
- 386 and the spatial resolution of emission estimates (20 km  $\times$  20 km vs. 3 km  $\times$  3 km). Our sensitivity

- analysis shows that changing the prior fluxes, meteorological field, and emission estimation resolution
  to roughly match Pisso et al. (2019) did not produce a result substantially different from the emission
  estimation result of the reference inversion. We thus concluded that the improved accuracy of emission
- 390 estimates in our study may be due to the use of columns as observational data. Column data are less
- 391 susceptible to the effect of PBL height changes that are difficult to simulate in transport models and
- 392 have information on a larger area of emissions due to the difference in wind direction at each altitude."
- 393
- L535: With "forward simulation," as pointed out above, how is footprint-based (backward is assumedunless explicitly stated) inversion possible?
- In the revised manuscript (Section 4.2), we have added an explanation that the forward simulations
  correspond to calculating the XCO2 values from the footprints and the surface CO2 fluxes using
  Equations (2) to (4).
- 399

L540: The mismatch between predictions and observations could be due to local sources not included 401 in the prior, not necessarily due to transport error. Do you have evidence that there was a clear transport 402 error? For CH4, EDGAR is generally not as good as regional inventories. I see both CO2 and CH4 403 measurements are significantly higher later in the afternoon (from Figure S3). It seems that the CO2 404 and CH4 sources are correlated. It may be the transport model didn't capture the afternoon winds. Any 405 evidence for that?

- We have no clear evidence to suggest that there was an error in the transport (e.g., wind speed and direction do not substantially differ from the measurements; simulated PBL heights do not take extreme values). On the other hand, as you pointed out, the mismatch between the prior simulations and the observation on 3 March 2016 may be attributable to a short-term local source not included in
- 410 the prior fluxes.
- 411 We have revised the sentence as follows: "As described in Sect. 4.2, in some cases, the simulations
- 412 failed to reproduce the diurnal variation and to capture the plume from nearby large point sources,
- 413 possibly because of the transport modeling error or the short-term local sources not included in the
- 414 prior fluxes (Figs. 8d and S6)."
- 415

Referee #2

The authors developed an inversion scheme to infer the anthropogenic carbon dioxide emissions in 418 the Tokyo Metropolitans Area from observations of three ground based remote sensing sites. One of 419 which is a TCCON site.

The authors obtained the background by subtracting the simulated CO2 enhancement (from the 422 footprint and surface flux) from the observed XCO2 values at the Tsukuba COCCON site for the 423 forward modeling. To assess the biosphere, they spatially downscaled the terrestrial biospheric model 424 VISITc to simulate the biogenic influence and found the influence to the enhancements to be small. 425 The authors infer the meteorological surface interaction using WRF-STILT with a spatial resolution 426 of 1km. The Bayesian inversion scheme inverts for spatially resolved emissions, separated into point 427 and area-sources for the more than two months period with a total of approximately 6.5 degrees of 428 freedom. The authors also compared 12 different model configurations.

The authors report total carbon dioxide emissions for the study area and compare it to several literature reports and find good agreements within the reported uncertainties. The scientific value is to be rated as high, since emission estimates from observations still remain a tough challenge and needed to confirm or refine reported emission inventories. The paper is written in a clear, structured style. However, some details need improvements.

We thank you for your careful reading of our paper and for providing many valuable comments. We
have added descriptions related to the potential weaknesses that you raise and revised our manuscript
according to your comments. Please see our specific responses below.

Potential weaknesses are:

1. The authors conduct inversion in log-space, and therefore negative emissions are suppressed, which is not very realistic. The biogenic model needs to be perfect, so that we can be sure that there are no"negative emissions".

This study does not optimize total (anthropogenic + biogenic) fluxes, but only anthropogenic fluxes.

Because the magnitude of the biogenic fluxes (negative fluxes during the daytime) in the Tokyo

Metropolitan Area (TMA) in February and March is more than an order of magnitude smaller than the anthropogenic fluxes and their differences among four models are small (with a standard deviation of

0.09 ppm), the biogenic fluxes were fixed at the prior values. Therefore, it is reasonable to constrain the anthropogenic fluxes (nonpoint or area sources) to positive values by the inversions in log-space.

In the revised manuscript, we have made it clear that only "anthropogenic" emissions are optimized.

- 452 2. DOFS of 6.49 implicates that solution tend to stick to the a-priori, given that the dimension of the 453 state vector is rather large (m = 1921 or 481 or 121).
- 454 As you suggested below, we have investigated how the degrees of freedom for signal (DOFS) change
- when the prior uncertainties are increased by a factor ~1.5 and 2. Although the DOFS increase with 456 the prior uncertainty, we found that their changes are not very large (please see our response below).
- 457 However, a DOFS of ~6.5 would be useful for evaluating emissions from administrative divisions. In
- 458 the present study, the focus was on emissions for each administrative division rather than smaller-scale
- 459 individual emissions, and we compared the estimated emissions aggregated with the administrative
- 460 boundaries with the reported administrative emissions.
- 461

3. The authors assumed that all the sites have the same background air. It is not always true, when 463 considering the transport time that the air needed to travel from upwind to downwind especially when 464 the distance between the sites are big ( $\sim 60$  km).

- 465 The background values used in the simulation and the inverse analysis are specified as the XCO2
- 466 measurements at Tsukuba minus the STILT-calculated  $XCO_2$  enhancements ( $\Delta XCO_2$ ) at Tsukuba.
- 467 These background values correspond to the concentrations at the boundary of the TMA defined in this
- 468 study, and we think it is appropriate to consider the background to be common to the observation sites
- 469 within the relatively small TMA. The XCO2 values for urban sites other than Tsukuba are represented
- 470 as the sum of the background and the  $\Delta XCO_2$  calculated in consideration of fluxes and atmospheric 471 transport within the TMA.
- 472
- 473 4. The definition of background is confusing. The authors have two definitions of background in the 474 paper, i.e. 5 percentile value of TCCON station at Tsukuba and observed XCO2 from Tsukuba 475 COCCON site subtracted with simulated CO2 enhancement.
- 476 In the revised manuscript, the 5-percentile value of the Tsukuba TCCON site has been referred to as 477
- the "baseline". The "background" is now used only in the simulations. We note that the Tsukuba
- 478 COCCON data were not used to estimate emissions but used only to correct XCO2 values observed
- 479 by the other spectrometers.
- 480
- 481 I would appreciate if the authors could comment on their thoughts on the potential weaknesses and/or 482 discuss it in the paper, before the acceptance.
- 483 We have added the discussion regarding the limitations and possible improvements from both the
- 484 measurement and simulation sides in Section 5 (L601-605 and L579-583, respectively).
- 485 Briefly stated, from the measurement side, one limitation is the number of measurements. More
- 486 instruments and longer time series would probably increase our sensitivity and thus the DOFS. More
- 487 instrument locations would also help to constrain the background. Another limitation from the

- 488 simulation side is the difficulty of accurately modeling the wind fields. As we saw for 3 March 2016,
- we had mismatches possibly due to imperfect wind fields. To better constrain wind fields and PBL,additional wind lidar observations would be useful.
- 491
- 492 Detailed comments:

L 15: Suggestion: "We conducted ..." --> "In order to infer a top down emission estimate, we 494 conducted..."

- 495 We have made this revision.
- 496

L17: I thought that you deployed 3 EM27SUN spectrometers, please clarify.

As described in Section 2, the SN63 EM27/SUN arrived in Tsukuba in the middle of the campaign, and sunlight measurements were not performed during the entire campaign period (i.e., only from March to April 2016). To avoid any misunderstanding that the three EM27/SUNs were used for emission estimates, we would like to keep this description here. For clarification, we have added the following sentence in Section 3.4: "In the following simulations and inverse analyses, only the TCCON data were used as the measurement data at Tsukuba, since the SN63 EM27/SUN measurements started in the middle of the campaign (as described in Sect. 2)."

- 507 We have made these revisions.
- 508

L 26: "emission fluxes at > 3km" To my understanding, the WRF-STILT resolution is 1km, please
clarify.

We have added the following description in Section 3.4: "For area source emissions, however, we re- gridded the original footprints to a spatial resolution of  $0.025^{\circ} \times 0.025^{\circ}$  to degrade the spatial resolution for the inverse analysis. First, the area source emissions were summed for each  $0.025^{\circ} \times$

$0.025^{\circ}$  grid cell. Then, individual footprints for the  $0.025^{\circ} \times 0.025^{\circ}$  grid were derived by dividing the

- 515 sum of the nine XCO2 contributions for the  $0.0083^{\circ} \times 0.0083^{\circ}$  grid by the emissions for the  $0.025^{\circ} \times$
- 516 0.025° grid."
- 517

L 31: Please add your final emission number for the study area, or at least the scaling factor with the 519 according uncertainty to the abstract and if feasible, compare it to the literature references.

We have revised the last two sentences of the abstract as follows: "The prior and posterior total CO2

- emissions in the TMA are  $1.026 \pm 0.116$  and  $1.037 \pm 0.054$  Mt-CO2 d-1 at the 95% confidence level,
- 522 respectively. The posterior total CO2 emissions agreed with emission inventories within the posterior
- 523 uncertainty, demonstrating that the EM27/SUN spectrometer data can constrain urban-scale monthly

L 22: "nonpoint source" --> I would suggest the term "area source" (29 occurrences)

| 524 | CO 2 emissions."                                                                                    |
|-----|----------------------------------------------------------------------------------------------------------------|
| 525 |                                                                                                                |
| 526 | L 87: Suggestion: "when the daily sunshine duration in this region is high"> "during the high-                 |
| 527 | insolation period" for clarity and specificity.                                                                |
| 528 | We have revised the sentence as follows: "when the proportion of clear days is high"                           |
| 529 |                                                                                                                |
| 530 | L 93: "city center"> "city-center"                                                                             |
| 531 | We have made this revision.                                                                                    |
| 532 |                                                                                                                |
| 533 | L101: "ASL"> "a.s.l." (standard abbreviation, multiple occurrences)                                            |
| 534 | We have made this revision.                                                                                    |
| 535 |                                                                                                                |
| 536 | L107: " and is now continuously operated "> " and has since been continuously operated "                       |
| 537 | We have made this revision.                                                                                    |
| 538 |                                                                                                                |
| 539 | L116: "interval of approximately 1 min"> "interval of about 1 minute"                                          |
| 540 | We have made this revision.                                                                                    |
| 541 |                                                                                                                |
| 542 | L130: What is the integration time for determining sigma? If it is 1 min, it might be useful to also           |
| 543 | report the 15 min values as you did for comparing the observations with the forward simulations.               |
| 544 | An integration time of 15 min was used. We have added the following sentence: "Each of the                     |
| 545 | EM27/SUN data points was averaged per 15-min bin."                                                             |
| 546 |                                                                                                                |
| 547 | L132: you scale the TCCON to EM27, would it not make more sense to scale EM27 to TCCON, since                  |
| 548 | TCCON is considered as standard.                                                                               |
| 549 | It is certainly common to scale EM27/SUN data to TCCON data. However, the Tsukuba TCCON                        |
| 550 | XCO 2 data have a slightly larger scatter than the other EM27/SUN data used in this study, and this |
| 551 | made the variation in the TCCON XCO 2 data at a high solar zenith angle somewhat ambiguous. To      |
| 552 | derive an airmass-dependent correction factor (ADCF) for the SN44 EM27/SUN, we used the SN63                   |
| 553 | EM27/SUN data as the reference, which were validated using co-located aircraft measurements                    |
| 554 | (Ohyama et al., 2020).                                                                                         |
| 555 | We note that, in analyses where measurements at one site are used as part of the background for                |
| 556 | measurements at other sites, the differences between them (enhancements above the background)                  |
| 557 | rather than the absolute values of the concentration are particularly important. Which instrument is           |
| 558 | used as the reference has little effect on the emission estimates. In fact, in the case where the SN44         |
| 559 | EM27/SUN XCO2 data corrected for the airmass dependence and the SN38 EM27/SUN XCO2 data                        |

| 560 | were scaled to the original TCCON data, the relative change in the total TMA CO 2 emissions is less      |
|-----|---------------------------------------------------------------------------------------------------------------------|
| 561 | than 0.1%.                                                                                                          |
| 562 |                                                                                                                     |
| 563 | Ohyama, H., Morino, I., Velazco, V. A., Klausner, T., Bagtasa, G., Kiel, M., Frey, M., Hori, A., Uchino,            |
| 564 | O., Matsunaga, T., Deutscher, N. M., DiGangi, J. P., Choi, Y., Diskin, G. S., Pusede, S. E., Fiehn, A.,             |
| 565 | Roiger, A., Lichtenstern, M., Schlager, H., Wang, P. K., Chou, C. CK., Andrés-Hernández, M. D.,                     |
| 566 | and Burrows, J. P.: Validation of $\mathrm{XCO}_2$ and $\mathrm{XCH}_4$ retrieved from a portable Fourier transform |
| 567 | spectrometer with those from in situ profiles from aircraft-borne instruments, Atmos. Meas. Tech., 13,              |
| 568 | 5149-5163, https://doi.org/10.5194/amt-13-5149-2020, 2020.                                                          |
| 569 |                                                                                                                     |
| 570 | L135: Maybe mention the altitudes of the stations somewhere in the text.                                            |
| 571 | The altitudes of each station are described in the first paragraph of Section 2.                                    |
| 572 |                                                                                                                     |
| 573 | L142: unit wrong $\rightarrow ppm/(mol/m2/s)$                                                                       |
| 574 | We have made this revision.                                                                                         |
| 575 |                                                                                                                     |
| 576 | L145: Since you use the exact same altitudes as T.S.Jones et al., 2021 uses, you can add a citation here.           |
| 577 | The paper by Jones et al. (2021) has been cited here.                                                               |
| 578 |                                                                                                                     |
| 579 | L154: "multiplied by anthropogenic and biogenic fluxes"> "multiplied with spatially resolved                        |
| 580 | emission inventories for anthropogenic and biogenic fluxes separately"                                              |
| 581 | We have made this revision.                                                                                         |
| 582 |                                                                                                                     |
| 583 | L155ff: "The change over all grid cells."> "The change over all grid cells and serves for the                       |
| 584 | forward modeling."                                                                                                  |
| 585 | We have made this revision.                                                                                         |
| 586 |                                                                                                                     |
| 587 | L150ff: "We then aggregated the footprints in each grid over the STILT run time." It is not clear what              |
| 588 | you mean by "aggregate". If it is meant as an introduction into the following sentence I would suggest              |
| 589 | to move the line break before this sentence.                                                                        |
| 590 | We have moved the position of the line break and revised the sentences as follows: "The hourly                      |
| 591 | footprints calculated over the STILT run time (24 h) at a given time were weighted by temporal                      |
| 592 | correction factors of $CO_2$ emissions (described in Sect. 3.3) and aggregated in each grid cell. From the          |
| 593 | summed footprints at each altitude, we then calculated the pressure-weighted column-average                         |
| 594 | footprint, taking account of the column-averaging kernel of the EM27/SUN spectrometer (Rodgers                      |
| 595 | and Connor, 2003; Jones et al., 2021)."                                                                             |

L243: please break down this long sentence into at least two shorter ones

We have revised the sentence as follows: "Specifically, hourly net ecosystem exchange (NEE) data from the Vegetation Integrative SImulator for Trace gases (VISIT) model, referred to as VISITc, were

- 600 adopted as the biogenic CO2 flux data. The NEE data were combined with green vegetation fraction
- 601 (GVF) data to downscale them."
- 602

L248: "the original VISIT" --> "the initial VISIT"

- 604 We have made this revision.
- 605

L251: Gaussian T382 Grid --> please explain shortly, give reference or just state something like
"operate on exactly the same grid" in order to make your point.

We have revised the sentence as follows: "The VISITc model operates on the same grid as the CFSR 609 data (i.e., approximately  $0.31^{\circ} \times 0.31^{\circ}$ )."

Section 3.3 in general: Multiple sentences are very long; consider breaking them into shorter pieces612 for a better understanding.

We have made this revision.

L257: How you can downscale VISITc product from 0.31 \* 0.31 deg. to 1 km x 1 km using 4 km
resolution GVF data? I am not sure whether you have the high-resolution information necessary to
achieve this goal.

The effective spatial resolution of the downscaled biogenic fluxes is about 4 km, although the biogenic 619 flux data were generated on a 1 km x 1 km grid to be consistent with the footprints. To avoid misunderstanding, we have revised the sentence as follows: "to better characterize the spatial distribution of biogenic CO2 fluxes, we spatially downscaled the hourly VISITc NEE data using GVF

data from the Visible Infrared Imaging Radiometer Suite (VIIRS) sensor onboard the Suomi National

Polar-orbiting Partnership satellite (VIIRS Global Green Vegetation Fraction). The GVF data are produced with an approximately 4-km spatial resolution on a daily basis from the past 7 days of VIIRS

observations (Ding and Zhu, 2018)."

- 626 In addition, we have added the following sentence: "We note that the effective spatial resolution of the
- 627 downscaled biogenic fluxes is about 4 km, although they were generated on a 1 km x 1 km grid."
- 628
- 629 L267: "DXCO2 values measured" This statement is confusing, since DXCO2 values are derived from
- 630 the forward model as described in the referenced section 3.1
- 631 In the revised manuscript, we have modified the forward model that calculates XCO2. We have revised

- 632 the sentence as follows: "XCO2 measurements at a given location are quantitatively related to the 633 presumed surface CO2 fluxes via the forward model H"
- 634
- 635 L268: "H, representing atmospheric transport" --> To my understanding it is the forward model.
- 636 We have removed "representing atmospheric transport".
- 637
- L273: Is it correct, that you have the logarithmic of a scaling factor (unitless) as well as an emission
  value (in mole/area/time) in the state vector x? Please clarify.
- 640 The scaling factor is linear, not logarithmic. We have added the following sentence: "On the other 641 hand, the scaling factor for the large point source emissions was optimized at linear scale."
- 642

L279: Inverting in the log-space introduces a strong bias to positive emissions. Negative emissions 644 are not necessarily non-physical, especially in case of CO2, because biospheric activity might be 645 stronger than assumed. Did you try to invert in linear space? Negative emissions could serve as a 646 sanity check here.

- 647 The inversion in linear space was only tried at the initial stage. In the present study, we do not estimate 648 total (anthropogenic + biogenic) fluxes, but only anthropogenic fluxes. The negative emissions for the 649 anthropogenic sources could cause large uncertainty in their emission estimates. In addition, in 650 February and March in the TMA, the magnitude of the biogenic fluxes (negative fluxes during the 651 daytime) is more than an order of magnitude smaller than the anthropogenic fluxes (Table 4 and Figure 652 S5) and their differences among four models are small (with a standard deviation of 0.09 ppm) (Section 653 4.2). Therefore, it is reasonable to constrain the anthropogenic fluxes to positive values by the 654 inversions in log-space.
- 655 In the revised manuscript, we have revised the sentence to make it clear that "anthropogenic area
- 656 source" emissions are optimized as follows: "because the area source emissions from each grid cell
- 657 differ by a couple of orders of magnitude, and the optimization of area source emissions at linear scale
- 658 might lead to unphysical negative posterior emissions."
- 659
- L331: "sources, large point sources" --> "sources, strong point sources" to separate from the spatial
   meaning of "large"
- In the revised manuscript, we have defined "large point sources" as point sources with large emissions(the first paragraph of Section 2).
- 664
- L311ff: It is not very clear in the text what XCO2{{Diff} means and how it separates from (DXCO2).
- 666 We have revised the sentence as follows: "To characterize the diurnal variation in XCO2 at each
- observation site, we examined the diurnal variation in XCO2 enhancements (XCO2Enh) above the daily

- 668 XCO2 baseline."
- Additionally, in the revised manuscript, "ΔXCO2" is used for only representing XCO2 enhancements
  calculated from the forward model.
- 671

L315: It is a bit confusing here, because you define another background (5 percentile value of the
Tsukuba TCCON site) than the one you use for the forward modeling. What XCO2{[Diff] is actually
used for? Just to look into temporal fluctuation?

- To avoid confusion with another "background" used in the simulations, the 5-percentile value of the Tsukuba TCCON site has been referred to as the "baseline" in the revised manuscript.  $XCO_2^{Enh}$ ( $XCO_2^{Diff}$  in the initial manuscript) values were calculated using only the observed  $XCO_2$  values to examine the temporal fluctuation at each site. These  $XCO_2^{Enh}$  confirmed that using  $XCO_2$ measurements at Tsukuba as background in the simulations would be valid (please see also the next response).
- 681

L312: Please explain the reasons to use Tsukuba as a background site.

- 683 We have added the following sentence in Section 3.4: "We assumed that the  $XCO_2$  values at Tsukuba 684 approximately represent background air, as there are lower  $CO_2$  emissions around Tsukuba (Fig. 3) 685 and the  $XCO_2$  values observed at Tsukuba were systematically lower than those at the other urban 686 sites, which can be seen from the  $XCO_2$  values in Fig. 2a."
- 687

L314ff and L355: How many days (or observations of the n=654 observations) were replaced byCarbonTracker?

We have added the following description: "For days when measurements at Tsukuba were notavailable (16, 17, 27, and 28 February and 23 March)"

L359: you averaged the data in 15 mins. Why is it optimal or in another word, why no drift of the 694 sensor is integrated? You could refer to: https://acp.copernicus.org/articles/16/8479/2016/acp-16-695 8479-2016.pdf, section 3.1, where the optimal integration time is determined by using Allan analysis. 696 We have added the following sentences in Section 2: "Chen et al. (2016) derive an optimal integration 697 time of 10 to 20 min, based on the Allan variance of two sets of EM27/SUN data from side-by-side 698 measurements. However, they used a shorter integration time of 5 min to derive the EM27/SUN 699 differences between upwind and downwind of local emission sources. In the present study, we found 700 that it is difficult for the XCO2 simulation to accurately reproduce the times at which point source 701 plumes are observed (Sect. 4.2), and a comparison of the simulations and observations at short time 702 intervals is not beneficial. Thus, we adopted an integration time of 15 min for the EM27/SUN data." 703

L370: You talked about the model-observation discrepancy, forward modeling vs. observation ismainly given by the errors in the WRF-STILT, what about the background error?

In this case, the background is represented as the Tsukuba TCCON XCO2 data minus the  $\Delta$ XCO2 simulations at Tsukuba (i.e.,  $XCO_2 \frac{Tsukuba}{TCCON} - \Delta XCO_2 \frac{Tsukuba}{STILT}$ ). If this background value in the late 707 708 afternoon became larger by ~4 ppm, the simulation would agree with the observation. Considering the uncertainty in the  $XCO_2 \frac{Tsukuba}{TCCON}$  data and the magnitude of the  $\Delta XCO_2 \frac{Tsukuba}{STILT}$  data (Figure S5c), we 709 710 believe that the effect of the background on the model-observation discrepancy would be small. 711 Meanwhile, as pointed out by Referee #1, short-term local sources not included in the prior fluxes 712 could contribute to the discrepancy. Therefore, we have revised the sentence as follows: "However, 713 we cannot rule out the possibility that short-term local sources not included in the prior fluxes may 714 cause the discrepancy between the prior simulations and the observations. Therefore, we attribute this 715 large model-observation discrepancy to errors in the WRF-STILT model, or to the short-term local 716 sources not included in the prior fluxes, or both."

L445: you are looking into the model-observation mismatch for the inverse modeling framework.

However, in your inversion you assume the same background for all sites. The background influence
is canceled out in the forward model. Why you need to take the uncertainty of the background into
account?

Indeed, Equation (7) in the initial manuscript seems to indicate that the background is canceled out.
In the revised manuscript, the equation has been modified to represent how the urban XCO2
measurements are simulated. Equations (2) to (4) in the revised manuscript make it clear that
background (i.e., Tsukuba TCCON minus Tsukuba STILT) is included in the simulation.

We note that this change is mathematically identical (with just a movement of the  $XCO_2 \frac{Tsukuba}{TCCON}$  term), resulting in the same inversion results.

L455: It is not exactly clear what the authors mean with "upward" and "downward"

- 730 We have revised the sentence as follows: "the emissions from the central TMA region became smaller
- than the prior values, and the emissions from the other regions became larger than the prior values."
- T33 L470: With 6.5 degrees of freedom the model has not enough freedom to scale the sources individually.
- 734 What happens if you provide an intentionally much uncertain a-priori (e.g. Factor 2 higher).
- 735 When the prior uncertainties are increased by a factor ~1.5 and 2 (i.e., 120% and 170% of the prior
- emissions, respectively), the degrees of freedom for signal (DOFS) are 8.35 and 10.18, respectively.
- 737 Although the DOFS increase with the prior uncertainty, they still seem insufficient to resolve the
- 738 sources individually. We note that the case with 120% uncertainty is included in the sensitivity analysis.
- 739 In addition, the DOFS for all sensitivity analyses have been added to Table 5.

L475ff: Table 5: Please add the degrees of freedom and the Bayesian Information Criterion (BIC) tothis list. The latter is a helpful number to tell which of the models could be a better choice.

The DOFS have been added to Table 5. In addition, we calculated the BIC according to Rayner (2020)

(Table R1). With coarser spatial resolution (cases #7a and #7b), the BIC becomes smaller (i.e., a better model) due to the substantial decrease in the  $m \log(n)$  term of the BIC. According to this parameter, the worse the spatial resolution, the better the inverse model. We acknowledge that there are a variety of ways to optimize the grids for spatially resolved emission flux estimates, and we intend to consider this in future studies.

Rayner, P.: Data assimilation using an ensemble of models: a hierarchical approach, Atmos. Chem.

Phys., 20, 3725–3737, https://doi.org/10.5194/acp-20-3725-2020, 2020.

Table R1. Bayesian information criterion (BIC) for the different meteorological data, prior emission data, prior uncertainty ( $\sigma_a$ ), spatial correlation length of  $S_a(l_s)$ , temporal correlation length of  $S_{\epsilon}(l_t)$ ,

| 755 | and spatial | resolution | of the | inversion | domain | $(r_{\rm s})$ |
|-----|-------------|------------|--------|-----------|--------|---------------|
|     | r           |            |        |           |        | (° 9)         |

| Case | Meteorological data + prior | $\sigma_{\mathrm{a}}$ (%) | l s (km) | $l_{t}(h)$ | $r_{\rm s}$ (°) | BIC   |
|------|-----------------------------|---------------------------|----------------------------|------------|-----------------|-------|
|      | emission data               |                           |                            |            |                 |       |
| #0   | WRF/MYJ + ODIAC             | 85                        | 10                         | 1          | 0.025           | 11932 |
| #1   | WRF/MYJ + ODIAC (LPS fixed) | 85                        | 10                         | 1          | 0.025           | 11958 |
| #2a  | WRF/MYNN25 + ODIAC          | 85                        | 10                         | 1          | 0.025           | 11912 |
| #2b  | WRF/YSU+topo + ODIAC        | 85                        | 10                         | 1          | 0.025           | 11890 |
| #2c  | ERA5 + ODIAC*               | 85                        | 10                         | 1          | 0.025           | 11719 |
| #3a  | WRF/MYJ + ODIAC             | 50                        | 10                         | 1          | 0.025           | 11937 |
| #3b  | WRF/MYJ + ODIAC             | 120                       | 10                         | 1          | 0.025           | 11927 |
| #4a  | WRF/MYJ + ODIAC             | 85                        | 5                          | 1          | 0.025           | 11933 |
| #4b  | WRF/MYJ + ODIAC             | 85                        | 20                         | 1          | 0.025           | 11932 |
| #5a  | WRF/MYJ + ODIAC             | 85                        | 10                         | 0.5        | 0.025           | 11854 |
| #5b  | WRF/MYJ + ODIAC             | 85                        | 10                         | 2          | 0.025           | 12161 |
| #6   | WRF/MYJ + EDGAR             | 95                        | 14                         | 1          | 0.025           | 12752 |
| #7a  | WRF/MYJ + ODIAC             | 75                        | 16                         | 1          | 0.05            | 3324  |
| #7b  | WRF/MYJ + ODIAC             | 65                        | 25                         | 1          | 0.1             | 1138  |

\*Data from Sodegaura on 23 March 2016 were excluded.

L494ff: The statement appears reasonable. However, referenced Fig. S6 does not appear to have a connection to this statement.

- 760 The reference to Figure S6 (Figure S7 in the revised manuscript) has been changed to the sentence
- describing EDGAR as follows: "case #5, EDGAR version 6 ( $0.1^{\circ} \times 0.1^{\circ}$  spatial resolution) without
- 762 large point source correction used as the prior estimate (Fig. S7)"
- 763
- 764 L497ff: The sentence is very long. Please reformulate.

We have revised the sentence as follows: "Although the number of grid cells with a spatial resolution
of 0.05° and 0.1° was equivalent to or lower than the number of measurement data points, respectively,
the total DOFS slightly decreased (to 5.84 for 0.05° and 5.05 for 0.1°). This was due to the changes in

- the prior uncertainty and the spatial correlation length."
- 769

L526: I thought a third EM27/SUN is also deployed at Tsukuba site.

As described in our response above, since the solar measurements with the SN63 EM27/SUN were not used for emission estimates, we would like to keep this description here. For clarification, we have added the following sentence in Section 3.4: "Note that in the following simulations and inverse analyses, only the TCCON data were used as the measurement data at Tsukuba, since the SN63 EM27/SUN measurements started in the middle of the campaign (as described in Sect. 2)."

L553: Again here is 3km resolution mentioned. To my understanding it is 1km. If not correct pleaseexplain the reasons.

We have added the following description in Section 3.4: "For area source emissions, however, we regridded the original footprints to a spatial resolution of  $0.025^{\circ} \times 0.025^{\circ}$  to degrade the spatial resolution for the inverse analysis. First, the area source emissions were summed for each  $0.025^{\circ} \times$  $0.025^{\circ}$  grid cell. Then, individual footprints for the  $0.025^{\circ} \times 0.025^{\circ}$  grid were derived by dividing the sum of the nine XCO2 contributions for the  $0.0083^{\circ} \times 0.0083^{\circ}$  grid by the emissions for the  $0.025^{\circ} \times$  $0.025^{\circ}$  grid."

- 785
- 786 L915: "sigma a" for prior uncertainty instead of "sigma e".
- 787 We have made this revision.
- 788

General model description:

lack of overview and strict separation of description of the inversion methodology, model setup detailsand results

- 792 We have refined the sentence structures in the revised manuscript. Specifically, we have moved the
- description of the simulation conditions in Section 4.2 (L331-342 in the initial manuscript) and the
- description of the construction of the prior error covariance matrix and measurement error covariance
- matrix in Section 4.2 (L387-412) to the Methodology section. Additionally, the descriptions of the

- background (L349-357 and L375-382) have been combined and moved to the Methodology section.
- The remaining part of Section 4.3 (L413-446) has been merged with Section 4.2, and Section 4.3 hasbeen removed.
- 799